# Cytohesin-2 is essential for the perinatal development of mice and regulates Golgi volume

Carsten Küsters[1],*, Bettina Jux[1],*, Farhad Shakeri[2,3], Sebastian Kallabis[4,5], Felix Meissner[4], Waldemar Kolanus[1]

Cytohesin proteins are guanine nucleotide exchange factors (GEFs) for ARF GTPases, particularly ARF1 and ARF6. Although *Arf1* and *Arf6* deficiency leads to embryonic lethality, the in vivo roles of cytohesins remain poorly characterized. In this study, we investigated the functions of cytohesin-2 both in vivo and in vitro. Strikingly, full knockout of cytohesin-2 in mice results in perinatal lethality within 20 h of birth. Employing mass spectrometry–based organellar proteomics for the cellular analysis of cytohesin-2 function, we discovered an altered Golgi apparatus in cytohesin-2–deficient C2 myoblasts. Specifically, immunofluorescence analysis demonstrated a significant reduction in Golgi volume compared with the control, which was restored by reintroduction of cytohesin-2 in an ARF-GEF–independent manner. Moreover, we discovered that canonical Golgi functions are impaired in cytohesin-2 deficiency: Peanut agglutinin staining showed a significant reduction in galactose/N-acetyl-galactosamine at the cell level. In addition, global protein secretion was markedly reduced in neonatal cytohesin-2 knockout mice, as determined by quantitative mass spectrometry–based proteomics. Together, our findings establish cytohesin-2 as an essential regulator of perinatal development and as a mediator of Golgi maintenance.

## Introduction

ADP-ribosylation factor (ARF) GTPases, depending on their cellular localization, are involved in processes such as membrane trafficking and organization of organelle structure (Gillingham & Munro, 2007). Embryonic lethality of *Arf1*- and *Arf6*-deficient mice proves their crucial function in the organism (Suzuki et al, 2006; Hayakawa et al, 2014). The regulation of ARF activity, consisting of a cycle of GDP to GTP exchange by guanine nucleotide exchange factors (GEFs) and GTP to GDP by GTPase-activating

proteins (GAPs), is therefore of importance as well. Although large GEFs for ARF GTPases such as GBP1 and BIG1/2 are mainly thought to be important for the organization of all parts of the Golgi apparatus, small GEFs are mainly located at the plasma membrane and endosomes. Proteins of the cytohesin (CYTH) family belong to the group of small GEFs and are therefore continuously discussed to be part of the vesicle transport machinery in the cell (Casanova, 2007; Gillingham & Munro, 2007).

The CYTH protein family consists of four members of which CYTH1 and CYTH4 are mainly found in leukocytes, whereas CYTH2 and CYTH3 appear to be ubiquitously expressed (Kolanus, 2007). Their protein structure consists of an N-terminal coiled-coil domain that allows interaction between CYTHs and other proteins. Furthermore, CYTH proteins can be recruited to the plasma membrane by binding to the phosphatidylinositol-phosphates PtdIns(4,5)P$_2$ and/or PtdIns(3,4,5)P$_3$ via their C-terminal PH domain. Of note is the existence of two isoforms, which differ in only one glycine residue within the PH domain. CYTH proteins in the two-glycine (2G) variant favor binding to PtdIns(3,4,5)P$_3$, whereas the three-glycine (3G) variant mainly binds to PtdIns(4,5)P$_2$. The GEF activity is mediated by the central sec7 domain (Chardin et al, 1996; Klarlund et al, 2000; Cronin et al, 2004). The CYTH function can be inhibited by the pan-CYTH inhibitor SecinH3 that binds to the sec7 domain and thereby blocks the GEF activity (Hafner et al, 2006). GEF-independent functions of CYTH proteins are barely recorded.

GEF activity of CYTH2 was mainly described for ARF1 and ARF6, and CYTH2 was found to regulate the cortical actin cytoskeleton in an ARF6-dependent manner, affecting epithelial cell migration (Santy & Casanova, 2001; Santy et al, 2005). In addition, CYTH2 was reported to regulate the endocytic route activating both ARF1 and ARF6, thereby facilitating Salmonella uptake, influenza virus transport, and membrane trafficking toward the autophagosome (Hurtado-Lorenzo et al, 2006; Humphreys et al, 2012; Moreau et al, 2012; Yi et al, 2022). CYTH2 was also implicated in growth factor receptor signaling, and its interaction with the Golgi apparatus was reported (Franco et al, 1998; Monier et al, 1998; Lee & Pohajdak,

[1]Department of Molecular Immune and Cell Biology, Life and Medical Sciences (LIMES) Institute, University of Bonn, Bonn, Germany   [2]Institute for Medical Biometry, Informatics and Epidemiology, Medical Faculty, University of Bonn, Bonn, Germany   [3]Institute for Genomic Statistics and Bioinformatics, Medical Faculty, University of Bonn, Bonn, Germany   [4]Systems Immunology and Proteomics, Institute of Innate Immunity, Medical Faculty, University of Bonn, Bonn, Germany   [5]Core Facility Translational Proteomics, Institute of Innate Immunity, Medical Faculty, University of Bonn, Bonn, Germany

Correspondence: wkolanus@uni-bonn.de
*Carsten Küsters and Bettina Jux contributed equally to this work

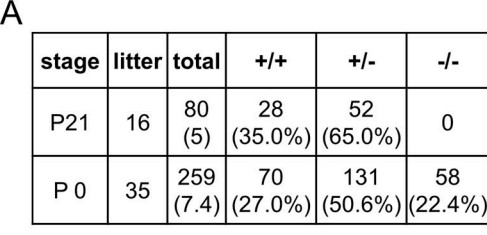

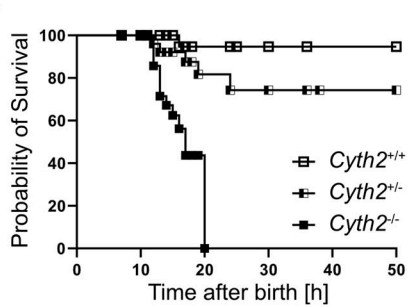

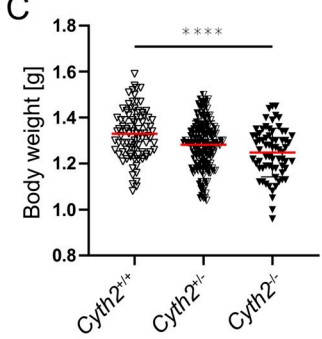

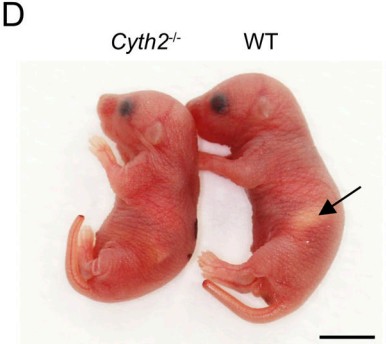

**Figure 1. *Cyth2* full knockout leads to neonatal lethality.**
**(A)** Percentage of viable *Cyth2* genotypes after weaning (P21) and of newborn mice (P0). Given is the number of litters and mice which were analyzed. **(B)** Survival rate of *Cyth2*[+/+], *Cyth2*[+/−], and *Cyth2*[−/−] mice. Time of death was estimated as the midpoint between two timed observations. A total of 115 neonates were analyzed (*Cyth2*[+/+] = 47 [1 died], *Cyth2*[+/−] = 54 [7 died], *Cyth2*[−/−] = 14). **(C)** Body weight analysis of newborn pups one to 2 h after birth. A total of 298 neonates were analyzed (*Cyth2*[+/+] = 82, *Cyth2*[+/−] = 151, *Cyth2*[−/−] = 65, one-way ANOVA, ****$P$ < 0.0001). **(D)** Image showing *Cyth2*[+/+] and *Cyth2*[−/−] mice 6 h after birth. The black arrow indicates the milk spot in the *Cyth2*[+/+], which is absent in the *Cyth2*[−/−] mouse. The scale bar represents 5 mm.

2000; Fuss et al, 2006; Hafner et al, 2006; Hurtado-Lorenzo et al, 2006; Jux et al, 2019). These findings derive from in vitro studies, and the only in vivo reports using conditional knockout mice for *Cyth2* attributed the small GEF to neuronal activity and eosinophilic inflammation (Yamauchi et al, 2012; Lolignier et al, 2015; Torii et al, 2015; Ito et al, 2021; London et al, 2022).

To analyze the global role of *Cyth2*, we have bred *Cyth2*-deficient mice, and we generated *Cyth2*-deficient cell lines by CRISPR/Cas9 to study CYTH2 function on vesicular transport in more detail in vitro. We found that all *Cyth2* full knockout mice die within 1 d after birth. Overall reduced protein secretion because of a disturbed Golgi volume and function might explain this severe phenotype. With this study, we document a thus far unappreciated function for CYTH2 in regulating the Golgi apparatus.

## Results

### *Cyth2* full knockout leads to neonatal lethality

In vivo functions of CYTH proteins are to date sparsely investigated; therefore, we bred full knockout mice of all four CYTH proteins. We and others observed no or minimal phenotypic effects in *Cyth1*, *Cyth3*, and *Cyth4* knockout mice at steady state (Yamauchi et al, 2012; Jux et al, 2019). In contrast, *Cyth2* full knockout mice (heterozygous mice with conditional potential were obtained from EMMA) from heterozygous breeding were not detectable after weaning by genotyping although they were born in a Mendelian ratio (Fig 1A). We observed that *Cyth2*[−/−] mice died between 12 and 20 h after birth (Fig 1B). One to 2 h after birth,

*Cyth2*[−/−] mice had significantly less weight of 5.5% compared with WT littermates (Fig 1C). Heterozygous littermates also had a slightly reduced body weight compared with WT mice, and some pups died earlier (7 of 54) (Fig 1B and C), but *Cyth2*[+/−] mice, which grew to adulthood, had no obvious phenotype. Besides the reduced weight, we found that *Cyth2*[−/−] mice never had a milk spot and were never observed to suckle (Fig 1D), which could explain the weight loss.

To identify the pathology of the severe, lethal phenotype of *Cyth2*-deficient mice, we aimed to investigate the role of *Cyth2* in various organs. *Cyth2* is ubiquitously expressed (Fig S1A), and the highest expression level among tested organs was found in the brain. We generated several tissue-specific *Cyth2* knockout mice; among others, we bred brain-specific (*Cyth2*[fl/fl] x nestin-Cre) and skeletal muscle-specific *Cyth2* knockout (*Cyth2*[fl/fl] x myogenin-Cre) mice. These tissue-specific knockout mice were vital and showed normal behavior without reduced lifespan (Fig S1B) and therefore excluded common neuromuscular defects as a reason for nonsuckling and death. Kidney failure and respiratory defects are potential reasons for neonatal lethality. Mice with kidney failure can survive often more than 24 h, and hematoxylin–eosin-stained cryosections from kidneys 6 h after birth looked inconspicuous (Fig S1C) (Turgeon & Meloche, 2009). Also, lung sections from *Cyth2*[−/−] mice looked comparable to WT controls in hematoxylin–eosin staining and severe lung defects usually lead to death within minutes (Fig S1D) (Turgeon & Meloche, 2009). However, although we observed normal breathing activity in knockout mice during the first hours after birth, *Cyth2*[−/−] pups developed cyanosis shortly before death. Thus, we cannot fully exclude respiratory defects as a cause of lethality. Furthermore, we generated heart-specific *Cyth2* knockout mice (*Cyth2*[fl/fl] x alpha-

myosin heavy chain-Cre, *αMyHC*-Cre), which also did not pheno-copy *Cyth2* full knockout mice (Fig S1B).

Disturbed glucose homeostasis is another potential cause of death at time points we observed (Turgeon & Meloche, 2009). Therefore, we generated and analyzed liver-specific (ls) knockout mice (*Cyth2*$^{fl/fl}$ x albumin-Cre, ls *Cyth2*$^{-/-}$). These mice were vital, with a normal lifespan, and showed no obvious abnormalities (Fig S2A). With food ad libitum, WT and ls *Cyth2*$^{-/-}$ mice had comparable weight and blood glucose level (BGL), normal expression of glucose homeostasis genes glucokinase (*Gck*), pyruvate kinase (*Pklr*), and phosphoenolpyruvate carboxykinase (*Pck1*), but a significant stronger expression of glucose-6-phosphatase (*G6pc1*) (Fig S2B–D). As insulin receptor signaling is decreased in *Cyth3*$^{-/-}$ mice (Jux et al, 2019), we tested whether this is also the case in ls *Cyth2*$^{-/-}$ mice. We found a significantly reduced phosphorylation of p70S6K (the ribosomal protein S6 kinase beta-1, also known as S6K, phosphorylates S6 ribosomal protein to induce protein synthesis) after 10 min of insulin injection without differences in phosphorylation of AKT (Fig S2E–G). Thus, the insulin receptor response in the absence of *Cyth2* is hampered as well; however, the misregulation appears at a different step of the signaling cascade compared with *Cyth3*$^{-/-}$ mice.

Downstream of AKT and upstream of p70S6K lies the kinase mTOR, which has previously been described as an essential sensor not only for growth factors, but also for amino acids and other metabolites (Saxton & Sabatini, 2017; Condon & Sabatini, 2019). Although insulin injections were not feasible with newborn mice, we still assessed the metabolic state of the *Cyth2*$^{-/-}$ mice. We analyzed BGL, serum amino acid level, and expression of glycolysis and gluconeogenesis genes in newborn *Cyth2*$^{-/-}$ mice (Fig 2). Although newborn *Cyth2*$^{-/-}$ mice were not suckling, they exhibited normal BGL compared with WT mice 6 h after birth (Fig 2A). The BGL is tightly regulated by glycolysis and gluconeogenesis; we therefore tested for the expression of glycolysis genes *Gck* and *Pklr* in neonatal liver. However, these mRNAs were barely detectable in WT and *Cyth2*$^{-/-}$ livers at 6 h of age. The expression of gluconeogenesis genes *G6pc1* and *Pck1* was significantly up-regulated in livers of *Cyth2*$^{-/-}$ mice compared with WT, which could explain the balanced BGL despite nonsuckling (Fig 2B).

It was described before that serum amino acid levels are decreased in neonatal mice with impaired autophagy and mTOR signaling (when blocking autophagy), respectively (Kuma et al, 2004; Komatsu et al, 2005). Also, CYTH2-ARF6 activity was reported to promote autophagosome formation (Moreau et al, 2012). We therefore wondered whether CYTH2 activity is required for autophagy and amino acid supply in newborn mice. Surprisingly, when measuring serum amino acids, we found increased levels in *Cyth2*$^{-/-}$ mice compared with WT (Fig 2C). Also, there was no indication for altered autophagy in *Cyth2*-deficient heart or lung tissue and observations in *Cyth2*$^{-/-}$ myoblasts indicated even mildly increased autophagic activity (Fig S3A–G). These observations speak against an autophagic defect in *Cyth2*-deficient mice or cells. Interestingly, when starved and resupplemented with FCS or amino acids, *Cyth2*$^{-/-}$ C2 myoblasts showed a significant reduction of mTOR-mediated induction of ULK1 phosphorylation, ULK1 being the central autophagy-inducing kinase (Fig S4A–C). Colocalization experiments for mTOR and LAMP2 for investigating the mTOR

recruitment to the lysosomal surface support a minor aberration in mTOR signaling, as *Cyth2*-deficient cells showed a small but significant reduction in colocalization (Fig S5) (Chen et al, 1985).

Taken together, *Cyth2* is essential for neonatal survival and involved in metabolic regulations. As metabolic changes were not severe and high plasma amino acid levels are not considered detrimental, we do not expect this to be the reason of neonatal death of *Cyth2*-deficient mice. Attempting to phenocopy the neonatal lethality of *Cyth2*-deficient mice, we observed increased expression of gluconeogenesis genes in hepatic tissue to be the only similarity between full knockout and liver-specific knockout mice. The nonreproducible full knockout phenotype in tissue-specific knockout mice leads to the conclusion that *Cyth2* might play a systemic, organism-wide role in early postnatal development.

## Organellar proteome maps reveal changes in Golgi structure

The neonatal lethality of full knockout mice, which could not be phenocopied by conditional knockout mice, and the knowledge that *Cyth2* is ubiquitously expressed and implicated in vesicle transport (Casanova, 2007) prompted us to ask whether a loss of *Cyth2* might hinder the correct transport and positioning of certain proteins such as growth factor receptors, the V-ATPase, or others (Fuss et al, 2006; Hafner et al, 2006; Hurtado-Lorenzo et al, 2006; Moreau et al, 2012; Yi et al, 2022). We chose an unbiased approach to answer this question and performed differential centrifugation of WT and *Cyth2*$^{-/-}$ C2 myoblasts to separate organelles by their sedimentation properties. The resulting fractions were subjected to mass spectrometry–based proteomics analysis, and the abundance of proteins in each fraction was compared between WT and KO to identify displaced proteins and altered organelles (Itzhak et al, 2019). Principal component analysis (PCA) separates different cellular compartments by specific marker proteins, previously described as organellar maps (Itzhak et al, 2016) (Fig 3A). Overall, the PCA plots were comparable between WT and *Cyth2*$^{-/-}$ cells, showing that overall cellular integrity is not disturbed by the knockout of *Cyth2*. We detected 177 proteins, which were significantly displaced in *Cyth2*$^{-/-}$ C2 myoblasts compared with WT cells in two independent experiments (Fig 3B, Table S1). These proteins clustered in two groups, and Fisher's exact test was performed to identify significantly enriched GO terms in these two clusters, among which we identified "signal-anchor" as the strongest enriched term (Fig 3C). Comprising many glycosylating enzymes, this term prompted us to analyze organellar markers, and indeed, a disturbed distribution of Golgi markers in *Cyth2*$^{-/-}$ cells compared with WT cells was visible with a clear displacement from the 5,500*g* fraction toward fractions of lower and higher centrifugation speed (Fig 3D and E). We also identified displacement of marker proteins of other cellular compartments, such as the plasma membrane (from the 5,500*g* fraction toward the 12,000*g* fraction), actin-binding proteins (ABP, from 3,000*g* toward fractions with higher centrifugation speed), and endosomes (from 5,500*g* fractions toward the 12,000*g* fraction), but to a lesser extent.

The localization of CYTH2 within a cell has not been fully elucidated so far because of a lack of specific antibodies. Different studies find the majority in the cytosol (Frank et al, 1998; Li et al,

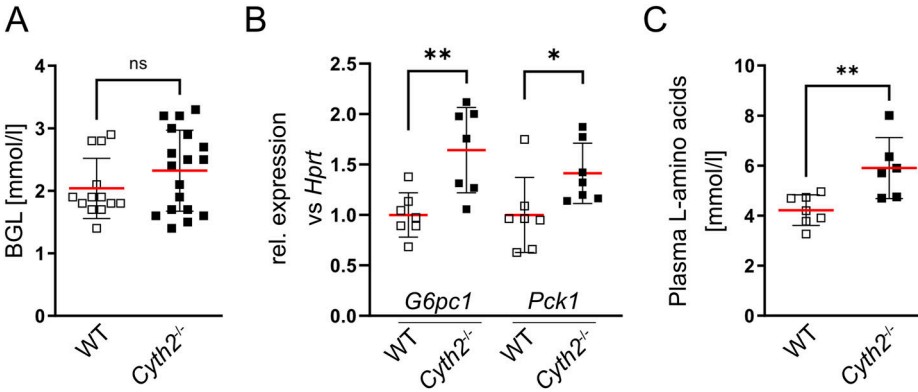

**Figure 2. *Cyth2* full knockout mice have increased gluconeogenesis and plasma amino acids.**
**(A)** Blood glucose levels 6 h after birth; samples were taken from the neck after euthanizing the mice, n = 13 (WT) and 18 (*Cyth2⁻/⁻*). **(B)** Gene expression of *G6pc1* and *Pck1* in livers from 6-hold WT and *Cyth2⁻/⁻* mice was analyzed by PCR. The expression was normalized to *Hprt*. **(C)** L-amino acids were measured in plasma isolated from 6-h-old mice, n = 6. Results are given in means ± SD. Every dot represents the result of a single mouse (*$P < 0.05$; **$P < 0.01$; ns, not significant).

2007), and at the plasma membrane (Frank et al, 1998; Venkateswarlu et al, 1998; Torii et al, 2010), endosomes (Maranda et al, 2001; Salem et al, 2015), and the Golgi compartment (Lee & Pohajdak, 2000). The localization might be dependent on the status of the cell. And the functions at different sites are not well understood but accepted for the endosomal pathway. Employing organellar maps, we found CYTH2 in all five fractions, showing that CYTH2 can localize to endosomes and the plasma membrane (fractions 5,500*g*–12,000*g*), but we also found a higher proportion of CYTH2 precipitating with large protein complexes (LPC, fraction 78,000*g*) (Fig 3F). We were further able to verify these mass spectrometric results by Western blot (Fig 3G).

Our organellar maps of *Cyth2⁻/⁻* C2 myoblasts compared with WT controls support a function for CYTH2 in vesicular transport of the endocytic pathway, because endosomal markers were displaced in *Cyth2*-deficient C2 myoblasts. In addition, the data indicate a novel role of CYTH2 in regulating Golgi-associated vesicle transport.

### Loss of *Cyth2* reduces Golgi volume in different cells from different species

To validate the findings from the protein localization analysis (Fig 3), we exploited immunofluorescence analysis. We confirmed changes in the endosomal compartment, as staining C2 myoblasts and A7r5 cells for RAB5 revealed increased numbers of early endosomes in *Cyth2⁻/⁻* cells compared with WT controls (Fig S6A–D). The accumulation of early endosomes was propagated to the late endosomal compartment as revealed by increased numbers of RAB7-positive structures in knockout cells (Fig S6A–D). In accordance, we observed elevated numbers of lysosomes (LAMP2-positive) earlier in *Cyth2⁻/⁻* C2 myoblasts compared with WT cells (Fig S3F). Although previous studies addressing a role of *Cyth2* in endocytosis focused on the localization of cargo proteins such as the transferrin receptor, we analyzed RAB5 and RAB7 as key regulators of endocytosis (Naslavsky & Caplan, 2023). In contrast to an endosomal phenotype, perturbation of the Golgi compartment as a consequence of *Cyth2* ablation was unexpected, and represents a novel aspect of CYTH2 biology, because the integrity of the Golgi compartment was so far mainly attributed to the function of large ARF-GEFs such as GBP1 and BIG1/2 (Casanova, 2007; Gillingham & Munro, 2007). CYTH2 was shown to have some

minor localization to the Golgi apparatus in Cos-1 cells before (Lee & Pohajdak, 2000), and it acts as a GEF for ARF1 (Chardin et al, 1996; Hafner et al, 2006; Cohen et al, 2007; Humphreys et al, 2012), but a regulation of the Golgi compartment by CYTH2 has not been explored. To validate morphological perturbation of the Golgi apparatus by ablation of CYTH2 function, we stained the Golgi complex in WT and *Cyth2⁻/⁻* C2 myoblasts with the common Golgi marker Golgin-97 (also Golga1), and evaluated the organelle structure using IMARIS software (representative pictures in Fig 4A). We found that overall Golgi volume was reduced in two independent CRISPR/Cas9 *Cyth2⁻/⁻* clones of C2 myoblasts compared with WT control cells (Fig 4B). To validate this further, we stained CRISPR/Cas9-generated *Cyth2⁻/⁻* HEK293T cells (human, kidney, Fig 4C) and A7r5 cells (rat, smooth muscle, Fig 4D) for Golgin-97. Comparable to C2 myoblasts, the deletion of *Cyth2* in HEK293T cells (Fig 4C) and A7r5 cells (Fig 4D) led to significantly reduced Golgi volumes. The *Cyth2* knockout was verified by Western blot (Fig 4C and D).

Taken together, CYTH2 is involved in the structural regulation of the Golgi apparatus in different cells and organisms.

### Golgi volume reduction can be rescued by CYTH2-3G in an ARF-GEF–independent fashion

CYTH2 is expressed in two isoforms, differing in a single glycine residue within the PH domain. To define which isoform is responsible for the observed regulation of Golgi volume, we first determined the *Cyth2* variant expressed in C2 myoblasts, A7r5 cells, and HEK293T cells. Sanger sequencing of amplified *Cyth2* transcripts isolated from WT cells revealed expression of both the 2G (~40%) and 3G isoform (~60%) (Fig 4E). This ratio was comparable between the different cell types analyzed. Therefore, rescue experiments were performed in C2 myoblasts using both isoforms. We furthermore included the GEF-inactive mutant (E156K; E|K) and a CYTH2 construct lacking the coiled-coil domain (Δcc), which should result in a reduced binding capacity to other proteins and recruitment to the Golgi apparatus (Lee & Pohajdak, 2000; Nevrivy et al, 2000; Mansour et al, 2002; Venkateswarlu, 2003). C2 myoblasts were transfected with fusion proteins of the mentioned CYTH2 variants (2G, 3G, E|K, Δcc) with RFP or an RFP-expressing vector alone as a control. Then, Golgin-97 was stained and the

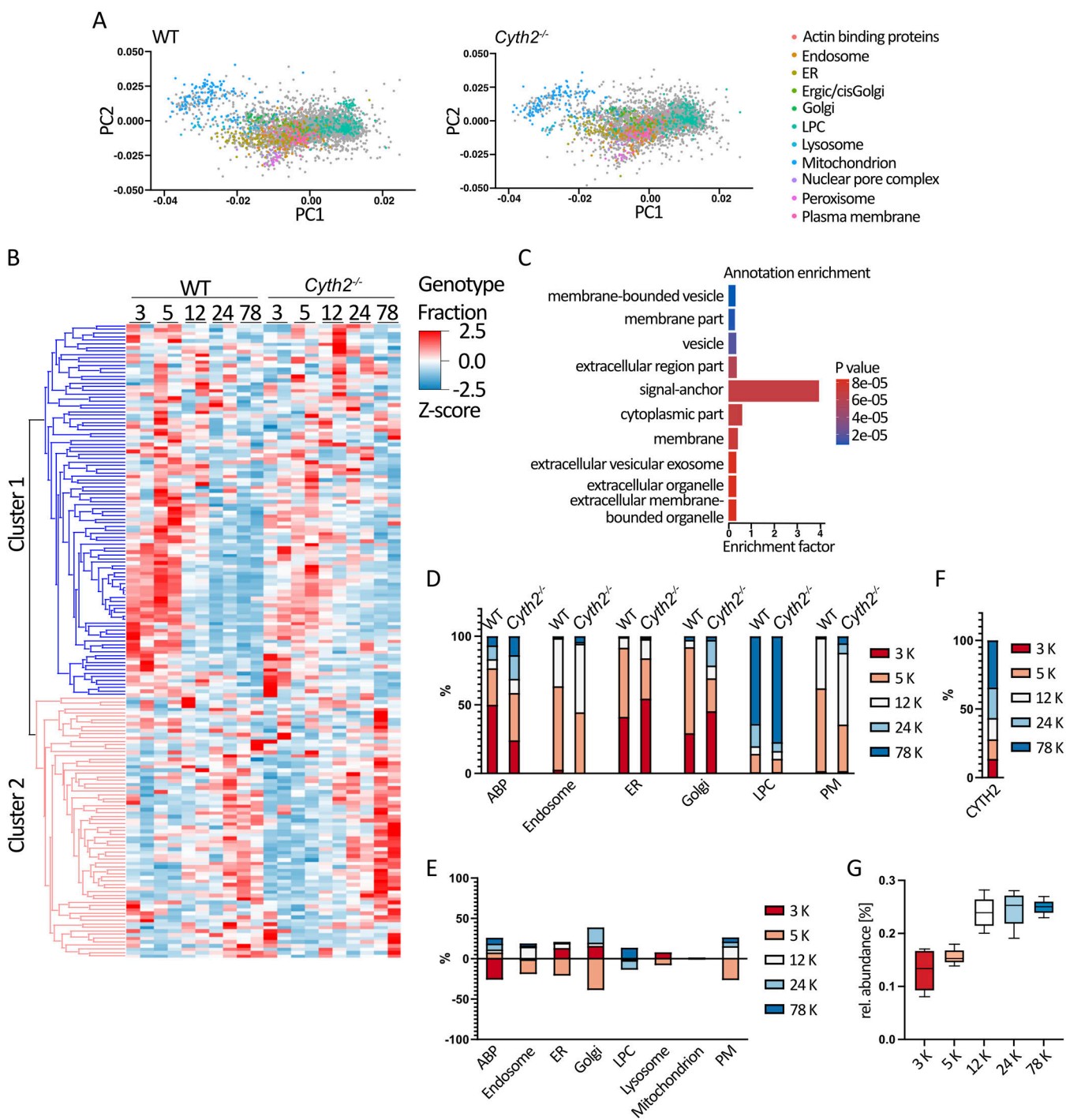

**Figure 3. Dynamic organellar proteome maps reveal a displacement in Golgi-associated proteins.**
**(A)** Representative PCAs of proteins detected in differential centrifugation fractions of WT (left) and *Cyth2*-deficient (right) myoblasts based on their LFQ intensities. Marker proteins as identified by Itzhak (Itzhak et al, 2016) are depicted in color according to the legend; gray dots represent all other proteins. **(B)** Heatmap of significantly displaced proteins overlapping in two independent experiments, assigned to two clusters (unsupervised). Depicted is the relative abundance of proteins as z-scores in the five differential centrifugation fractions (indicated by the number on top of each column, which represents the centrifugation condition of that fraction). **(B, C)** Fisher's exact test for GO-term enrichment in the two clusters in (B). **(A, D)** Percentage of major organellar proteins (as in (A)) across the five fractions, exemplary shown for actin-binding proteins (ABP), endosomes, the ER, Golgi, large protein complexes (LPC), and the plasma membrane (PM). **(D, E)** WT-KO difference of percentages of marker proteins in (D). **(F)** Distribution of CYTH2 protein over the five fractions determined by mass spectrometry. **(G)** Distribution of CYTH2 protein over the five fractions determined by Western blot. Data are presented as median with 75% quantiles for n ≥ 5 replicates.

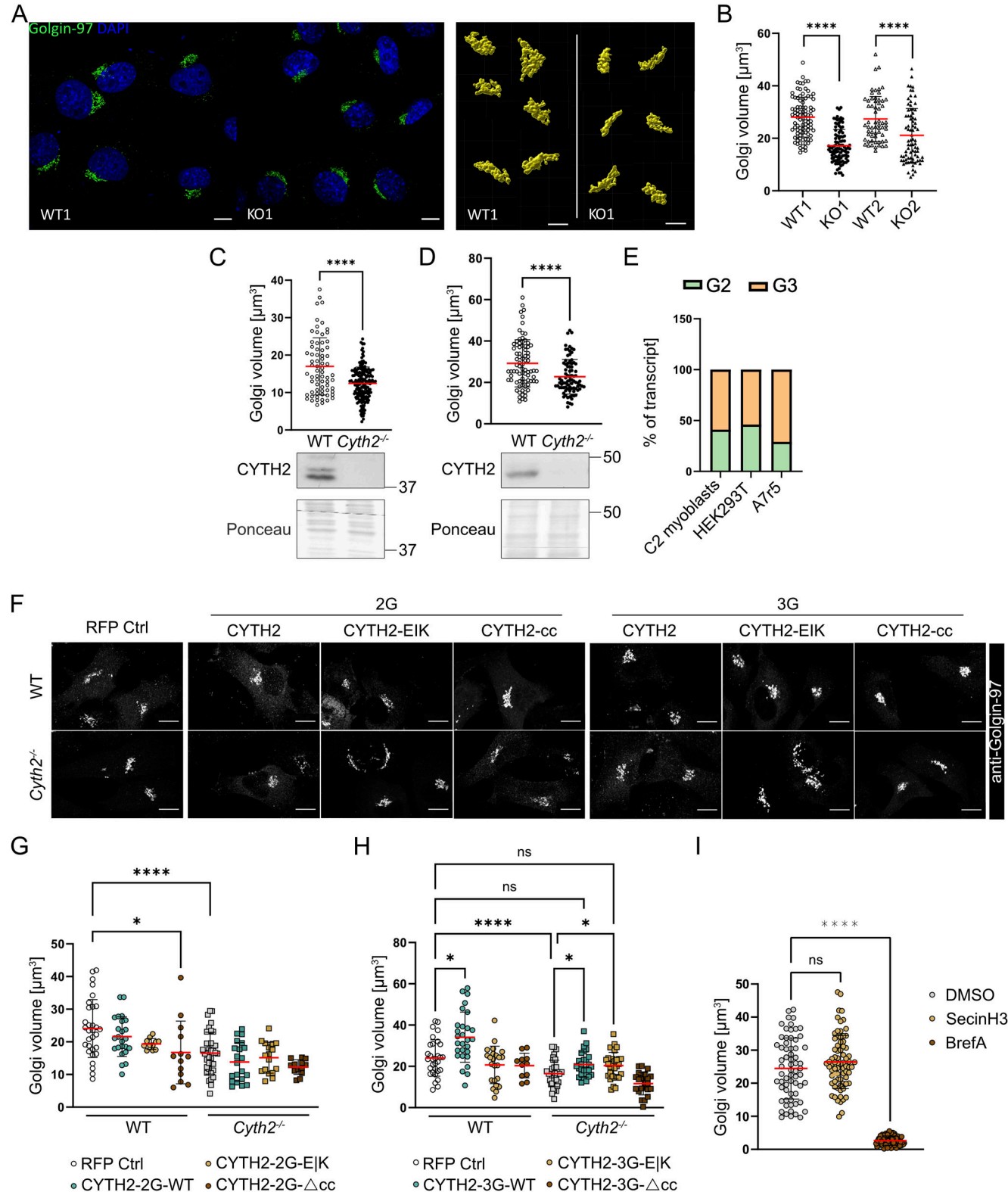

**Figure 4. Loss of *Cyth2* reduces Golgi volume in different cells from different species and can be rescued by CYTH2 overexpression.**
**(A)** Exemplary images of Golgin-97 immunofluorescence staining (left; scale bar: 10 μm) in WT and *Cyth2*-deficient myoblasts, from which confocal z-stack images were taken to reconstruct Golgi apparatus with IMARIS 9 surface detection (right; scale bar: 7 μm). **(B, C, D)** Golgi volume analyzed by IMARIS 9. **(B, C, D)** Two pairs of WT and *Cyth2*-knockout C2 myoblast clones, and (C) WT and *Cyth2*-deficient HEK293T and (D) A7r5 cells under standard culture conditions. **(C, D)** include Western blot controls for the depletion of *Cyth2*. **(E)** Quantified expression of the *Cyth2* 2G and 3G isoforms by Sanger sequencing and Tide analysis in WT C2 myoblasts, and HEK293T and A7r5 cells. **(F)** Exemplary immunofluorescence images of Golgin-97 in cells transfected with RFP or CYTH2-RFP fusion constructs. Scale bar represents 10 μm.

structure of the Golgi apparatus was analyzed by fluorescence microscopy in RFP-positive cells. RFP-transfected $Cyth2^{-/-}$ myoblasts showed decreased Golgi volumes compared with RFP-expressing WT controls (Fig 4F–H), confirming previous results and showing that the transfection method has no effect on Golgi volume. The expression of the 2G variant of CYTH2 could not reverse the decreased Golgi volume in $Cyth2^{-/-}$ C2 myoblasts, neither the WT nor the CYTH2-2G-E156K mutant or the CYTH2-2G-Δcc variant (Fig 4F and G). In contrast, introduction of WT CYTH2-3G in $Cyth2^{-/-}$ C2 cells was able to rescue the reduced Golgi volume (Fig 4F and H). Also, in WT C2 cells CYTH2-3G increased the Golgi volume over the levels of RFP-transfected WT cells. GEF-inactive CYTH2-3G (E|K) likewise rescued Golgi volumes in knockout myoblasts, whereas the CYTH2-3G-Δcc variant was unable to do so. Treatment of WT C2 cells with SecinH3, a small molecule inhibitor of the CYTH GEF function, did not change Golgi volume in contrast to brefeldin A, which caused disassembly of the Golgi complex (Fig 4I). This finding supports that the effect of CYTH2 on the Golgi volume is independent of its GEF function, at least in this in vitro system.

Taken together, these data show a clear involvement of CYTH2-3G in Golgi volume regulation, whereas the 2G variant has no effect on Golgi volume. As CYTH2-3G (E|K) expression was able to rescue the Golgi volume and SecinH3 treatment had no effect, we conclude that CYTH2 acts independent of ARF activation on Golgi regulation but might depend on binding to other proteins via its coiled-coil domain.

### *Cyth2* deficiency reduces Golgi functions such as glycosylation and secretion

The Golgi apparatus plays a major role in posttranslational modification of transmembrane and secreted proteins, involving the attachment of sugar moieties to the protein known as glycosylation (Marth & Grewal, 2008; Joshi et al, 2018; Schjoldager et al, 2020). These sugar residues form a broad range of antigens on proteins with different complexity, which are recognized by lectins, making lectins a biochemical tool to quantify these glycosylated residues. We therefore analyzed the glycosylation status of WT and $Cyth2^{-/-}$ C2 myoblasts, labeled cells with fluorophore-coupled lectin, and subjected them to flow cytometric analysis (Fig 5A). We stained with mannose-binding lectin (concanavalin A, ConA), galactose/N-acetyl-galactosamine–binding lectin (peanut agglutinin, PNA), and N-acetyl-glucosamine–binding lectin (wheat germ agglutinin, WGA). In all cases, cell surface tracing of protein glycosylation was significantly reduced by treatment of WT C2 cells with the Golgi inhibitor brefeldin A, showing that these glycosylation patterns are regulated by the Golgi apparatus (Fig 5A and B). Although binding of ConA and WGA was unaffected by *Cyth2* ablation, PNA

labeling was significantly reduced by 40% in two independent CRISPR/Cas9-generated $Cyth2^{-/-}$ C2 cell clones (Fig 5A and B), clearly showing a defect during the glycosylation process within the Golgi. This result was supported by the finding that the C1GALT1-C1GALT1C1 complex mediating galactose/N-acetyl-galactosamine modifications is displaced in the absence of CYTH2 as determined in the organellar maps analysis (Fig 5C).

As we here and others show, a reduced Golgi function leads to a false or reduced glycosylation (Misumi et al, 1986; Merk et al, 2009; Ignashkova et al, 2017; Ahat et al, 2022). This in turn could lead to a failure in the protein secretion process. Therefore, we asked whether we find fewer secreted proteins in the plasma of neonatal $Cyth2^{-/-}$ mice compared with WT mice (Table S2). We analyzed plasma samples from 6-h-old mice by mass spectrometry–based proteomics. The GO-term enrichment analysis of down-regulated proteins revealed terms of glycosylation and secretion as significantly enriched (Fig 5D). The volcano plot highlights that secreted proteins (colored in blue and red) are overall reduced in the plasma of $Cyth2^{-/-}$ mice (Fig 5E).

One of the severely reduced proteins in neonatal plasma of knockout animals was WFDC2 (WAP four-disulfide core domain protein 2). WFDC2 is a secreted glycoprotein, and it was recently shown that *Wfdc2* deficiency leads to neonatal lethality (Kirchhoff et al, 1991; Drapkin et al, 2005; Nakajima et al, 2019; Zhang et al, 2020). We therefore examined WFDC2 in plasma from 6-h-old mice by Western blot and confirmed the proteomics findings: in plasma from $Cyth2^{-/-}$ mice, WFDC2 was significantly reduced compared with plasma from WT and heterozygous mice. In contrast, in neonatal brains the expression level was comparable between WT and knockout mice, highlighting an effect of CYTH2 on WFDC2 secretion, not its expression (Fig 5F). Of note, because of its glycosylation secreted WFDC2 in the plasma has a higher molecular weight of ~27 kD compared with WFDC2 that resides in the tissue with a molecular weight of ~18 kD (Drapkin et al, 2005). As $Cyth2^{-/-}$ mice do not entirely phenocopy $Wfdc2^{-/-}$ mice, we assume that neonatal death of $Cyth2^{-/-}$ mice is attributed to diminished secretion of several proteins.

Taken together, these data indicate that CYTH2 regulates Golgi volume and function in various cell types. The absence of CYTH2 leads to impaired glycosylation and a failure in protein secretion.

## Discussion

In our organellar map analysis, we found a displacement of many proteins, among them ~20% of endosomal marker proteins. We confirmed changes in the endo/lysosomal compartment, finding the numbers of early and late endosomes, as well as lysosomes,

---

**(B, C, D, G, H)** Golgi volume analyzed as in (B, C, D) in WT and *Cyth2* knockout myoblasts overexpressing CYTH2 WT, E|K, and Δcc in the (G) 2G and (H) 3G isoform, with RFP serving as a control. **(B, C, D, I)** Golgi volume analyzed as in (B, C, D) in WT C2 cells treated with 50 μM SecinH3 or 5 μg/ml brefeldin A for 4 h, with DMSO serving as a control. Each symbol represents one Golgi apparatus analyzed; red lines indicate means ± SD (pooled data from more than three independent experiments). **(B, C, D, G, H, I)** Significance was tested using the Mann–Whitney test (B, C, D), two-way ANOVA (G, H, I), or one-way ANOVA (I). In (G, H), only relevant and significant comparisons are depicted (*$P < 0.05$; **$P < 0.01$; ***$P < 0.001$; ****$P < 0.0001$; ns, not significant). Source data are available for this figure.

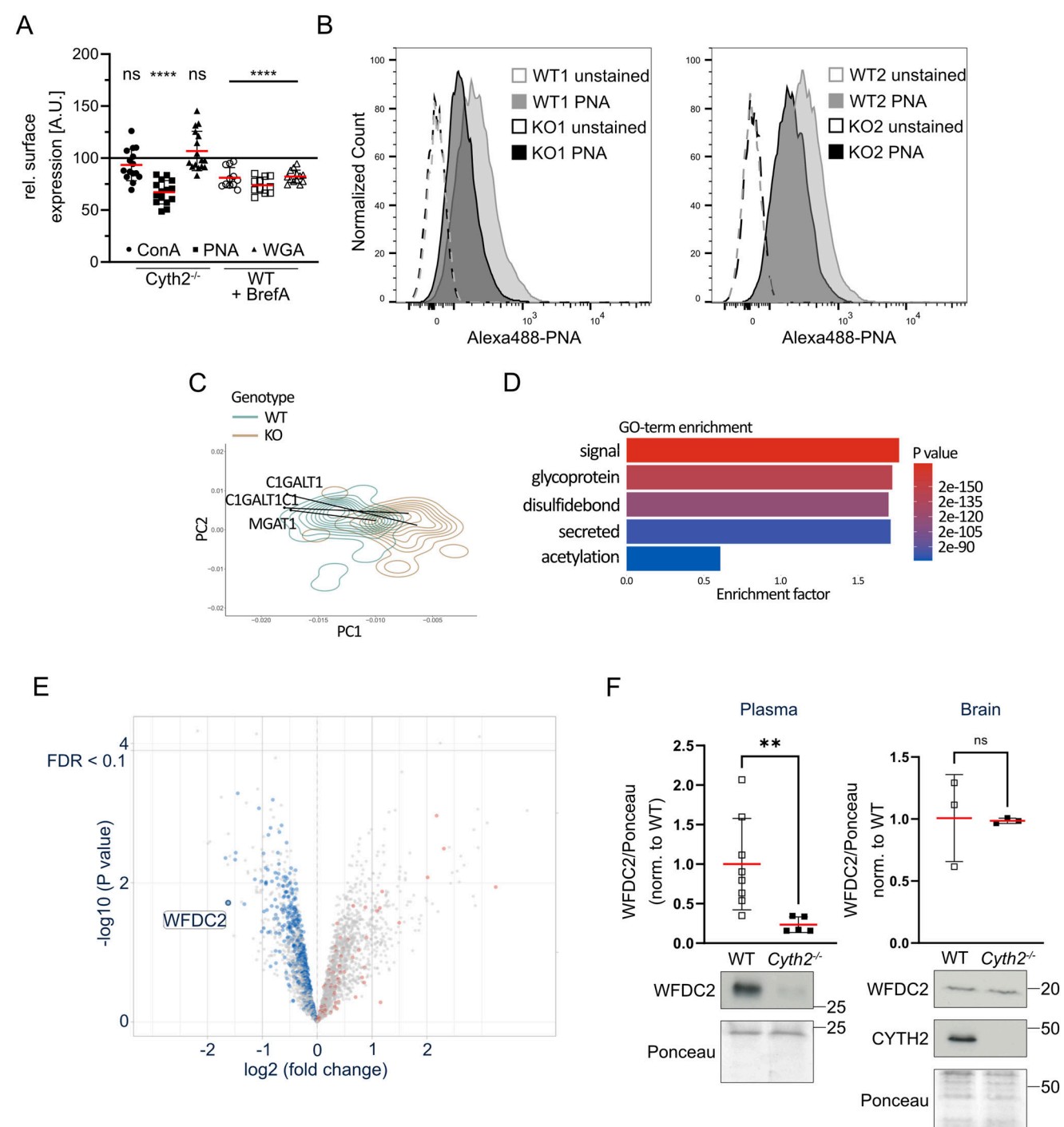

**Figure 5. *Cyth2^{-/-}* cells and mice show reduced galactose/N-acetyl-galactosamine glycosylation and impaired protein secretion, respectively.**
**(A)** Relative MFI of lectin staining (concanavalin A = ConA, peanut agglutinin = PNA, wheat germ agglutinin = WGA) of C2 myoblasts comparing *Cyth2*-deficient and brefeldin A–treated WT cells (5 µg/ml, 4 h) with untreated WT controls (100% reference line), measured by flow cytometry. Each dot represents one independent experiment (n ≥ 6); red lines represent the mean ± SD. **(A, B)** Representative histograms of PNA measured in two pairs of C2 myoblast clones as in (A). **(C)** Principal component analysis (PCA)–based density plots of the Golgi compartment (extracted from Fig 3A) overlaid with principal components (PC) 1 and 2 of C1GALT1, C1GALT1C1, and MGAT1 in WT (black dot) and *Cyth2* knockout (end of black line) myoblasts. **(D)** GO-term enrichment analysis of proteins measured in plasma from neonatal WT and *Cyth2*-deficient mice by proteomics (Fisher's exact test for down-regulated proteins). **(D, E)** Volcano plot of proteomics data analyzed in (D), comparing KO/WT. Secreted proteins (GO: KW-0964) are highlighted in blue (down) and red (up). All other proteins are represented as gray dots. **(F)** Western blot analysis of WFDC2 in neonatal plasma (left) or brain (right) from WT and *Cyth2*-deficient mice. Top: average signal intensity (red line ± SD; n ≥ 3) normalized to total protein loaded as determined by Ponceau staining. Bottom: representative immunoblot analyses are shown. Significance was tested with the Mann–Whitney test (*$P < 0.05$; **$P < 0.01$; ***$P < 0.001$; ns, not significant).

increased in cells lacking CYTH2 (Figs S3G and S6A–D). Our results shed new light on aspects of CYTH2 regulating the endocytic process, because previous studies had analyzed CYTH2-dependent endocytosis focusing on cargo protein transport alone, but data on the functional interaction of CYTH2 with regulatory proteins such as the RAB GTPases are lacking. The range of studies addressing varying aspects of endocytosis and a regulatory role of CYTH2 in the process emphasize the need for a comprehensive study of CYTH2-dependent endocytosis, observing cargo, regulatory proteins, and effector proteins at the same time (Hurtado-Lorenzo et al, 2006; Casanova, 2007; Gillingham & Munro, 2007; Humphreys et al, 2012; Moreau et al, 2012; Yi et al, 2022).

The unbiased analysis of subcellular protein localizations in *Cyth2*-deficient cells has revealed a novel aspect of *Cyth2* biology and identified many Golgi-associated proteins to be displaced (Fig 3). Furthermore, we could subsequently confirm changes in Golgi structure via immunofluorescence analysis, and found the Golgi volume to be significantly reduced in $Cyth2^{-/-}$ C2 myoblasts, and A7r5 and HEK293T cells compared with WT controls (Fig 4). Moreover, the Golgi volume could specifically be rescued by introducing CYTH2 into knockout cells. CYTH2 has GEF activity for ARF1, the major ARF GTPase regulating Golgi function (Gillingham & Munro, 2007; Sztul et al, 2019; Pennauer et al, 2022), but based on the rescue with the GEF-silent mutant, the effect of CYTH2 on Golgi volume in C2 cells appears to be GEF-independent (Fig 4H, E156K). Possibly, CYTH2 plays a role as an adaptor protein, whereas, for instance, the large GEFs activate ARF1 (Casanova, 2007; Gillingham & Munro, 2007). In addition, based on cell-free transport assays, an ARF-independent transport between Golgi cisternae was proposed (Happe & Weidman, 1998). Therefore, it seems more likely that CYTH2 affects the Golgi apparatus influencing the localization of proteins possibly because of interaction with the coiled-coil domain. This is underlined by the finding that the effect on Golgi volume could not be rescued by a cc-deficient CYTH2 mutant. The fact that CYTH2 was able to rescue the Golgi volume when specifically overexpressed in the three-glycine isoform points to the relevance of PtdIns(4,5)$P_2$ in this context and emphasizes the validity of our findings. As described before, "PtdIns(4,5)$P_2$ may contribute to the structural organization of the Golgi by interacting with cytoskeletal elements" (Mayinger, 2009).

Golgi architecture and integrity are essential for posttranslational modifications and transport of proteins throughout the cell, including protein secretion. Activation of ARF1 by ARF-GEFs plays an essential role in cargo sorting and vesicle formation (Arakel & Schwappach, 2018; Adarska et al, 2021). Although large GEFs such as GBF1 and BIG1/2 are essential to maintain Golgi integrity and function, small GEFs with a C-terminal PH domain such as CYTHs are described to act at the plasma membrane and endosomes (Casanova, 2007; Gillingham & Munro, 2007; Sztul et al, 2019). Even though a localization of CYTH2 to the Golgi apparatus depending on the coiled-coil domain was shown before (Lee & Pohajdak, 2000), investigations about its function at the Golgi are missing.

The present study suggests a structural and functional impact of CYTH2 on the Golgi apparatus in several cell types. The observed changes in Golgi volume could result from a decreased material influx or increased efflux from the Golgi, as size variation of this organelle is often caused by transportation defects (Guo &

Linstedt, 2006; Sengupta & Linstedt, 2011). CYTH2, an in vitro activator of ARF1, could directly or indirectly affect Golgi transport processes. It remains unclear, where in the cell CYTH2 acts to control Golgi size, whether it is at the Golgi apparatus itself or a remote function.

Failures in Golgi assembly are the cause of many diseases, and knockout of Golgi proteins leads to diverse phenotypes, ranging from early lethality to infertility and others (Zahn et al, 2006; McGee et al, 2017; Jiang et al, 2020; Kim et al, 2020). We found fewer secreted proteins in the plasma from $Cyth2^{-/-}$ mice compared with WT mice, which were not attributable to specific organs. One of the proteins was WFDC2. Recently, it was shown that $Wfdc2^{-/-}$ mice die 10 h after birth because of a lung failure. CYTH2 is highly expressed in the lung, and we cannot exclude lung defects as a cause of death of *Cyth2* full knockout mice. However, as $Cyth2^{-/-}$ mice do not entirely phenocopy $Wfdc2^{-/-}$ mice, we assume that neonatal death of $Cyth2^{-/-}$ mice is potentially attributed to overall reduced Golgi functionality, including diminished secretion of various proteins, reduced glycosylation, and disturbed localization of Golgi-passing proteins.

Among the displaced proteins in the organellar maps were both Golgi-associated proteins (such as C1GALT1, GALNT2, and B4GALT7), and proteins passing the Golgi on their way along the secretory pathway to the plasma membrane (the amino acid transporter SLC7A5). Although these instances are valuable hints for a Golgi defect, they remain single instances and they need to be interpreted carefully. However, reduced PNA binding on the surface of $Cyth2^{-/-}$ C2 myoblasts compared with WT cells indirectly confirmed the mislocalization of C1GALT1 and its functional consequence. Interestingly, also ULK1 was strongly mislocated in the organellar maps, whereas other mTORC1 components were less affected, reflected in the imbalanced mTOR response to amino acids and FCS. How and if the metabolic imbalance and ULK1 mislocalization are connected to the observed Golgi defect is unclear, yet ULK1 was shown to regulate Sec16a ER-to-Golgi transport and is therefore in functional vicinity to ARF1.

So far, only three in vivo studies have analyzed conditional knockout mice of *Cyth2*. Brain-specific $Cyth2^{-/-}$ mice (Ito et al, 2021) show reduced mechanical allodynia in inflammatory and neuropathic pain models, but otherwise, these mice were normal in lifespan. This confirms our observation that although CYTH2 is highest expressed in the brain, brain-specific knockout mice show no phenotype without challenge. Also, a Schwann cell–specific knockout of *Cyth2* causes no aberrant phenotype despite leading to a weak myelination defect (Torii et al, 2015). The most recent publication describes a reduced eosinophilic inflammation in conditional *Cyth2* mice, which were crossed with a germ cell–specific Cre line (London et al, 2022). However, it cannot be excluded that the knockout via Cre-recombination was not fully efficient and controls are missing, which might explain a rather mild phenotype of that particular mouse line. The $Cyth2^{-/-}$ mice of our study are indeed *Cyth2* full knockout mice as has been proven by Western blot (Fig S1A). Apart from reduced body weight, *Cyth2* knockout mice show normal appearance, movement, breathing, and no malformations. By breeding tissue-specific knockout mice, we exclude, for instance, neuromuscular or hepatic defects as a cause of death. Also, nonsuckling seems unlikely to be a direct

effect of the *Cyth2* knockout, but a secondary effect, for example, because of a general weakness deduced from the reduced weight of *Cyth2*$^{-/-}$ mice directly after birth. Our observations with various tissue-specific knockout mice emphasize that only a full knockout of *Cyth2* seems to have a severe effect and hints to a more systemic defect in full knockout animals.

Heart-specific *Cyth2* deficiency is no exception, yet needs to be mentioned here: cardiomyocyte-specific *Cyth2*$^{-/-}$ mice died 8–12 mo after birth because of a dilated cardiac hypertrophy (Fig S7A–F). However, it turned out to be an effect of the αMyHC-Cre line and seemed to be independent of the *Cyth2* expression (Fig S7G). The αMyHC-Cre line (Agah et al, 1997) is therefore not recommended to generate a stable heart-specific knockout (Li et al, 2023).

In the present study, we first describe that *Cyth2* deficiency leads to a reduced Golgi volume in different cell lines with functional consequences for glycosylation, protein transport to the plasma membrane, and potentially protein secretion in mice. The latter provides a plausible explanation for the neonatal lethal phenotype of *Cyth2* full knockout mice, yet requiring further investigation. Together, we could show that CYTH2 contributes to Golgi structural and functional integrity and is essential for the survival of mice.

# Materials and Methods

### Mice

*Cyth2*$^{+/-}$ mice (B6NDen;B6N-Cyth2$^{tm1a(EUCOMM)Wtsi/Ibcm}$) were obtained from European Mouse Mutant Archives (EMMA). Male and female C57BL/6N WT and *Cyth2*$^{-/-}$ mice originate from heterozygous breeding pairs and were bred under specific pathogen–free conditions at the animal facility of the University of Bonn. Mice were euthanized by decapitation at the age of 6 h for removal of organs and plasma. Animal care and experiments were performed according to German and institutional guidelines for animal experimentation and were approved by the government of North Rhine-Westphalia.

### Genotyping

Genotyping of neonatal mice was done from the tail tip. For traceability, mice were marked on the back with a number, which had no impact on the maternal behavior. DNA was isolated by incubation of tissue material with 200 µl 50 mM NaOH for 20 min at 95°C. The reaction was stopped by adding 70 µl Tris–HCl, pH 8, and subsequent centrifugation at 1,500g for 4 min. Primer sequences are given in Table 1.

### Blood sample handling

Blood samples were taken from neck after decapitation of neonatal mice or from the tail tip of adult mice, and blood glucose was determined with the Accu-Chek system (Roche). 20 µl of blood was collected in a heparinized Microvette CB300 (Sarstedt). After centrifugation, the plasma was transferred into a fresh tube and

stored at −80°C. For determination of plasma L-amino acids, the L-Amino Acid Quantification Kit from Merck was used according to the manufacturer's instruction.

### qRT-PCR

Total RNA was isolated with TRIzol and transcribed into cDNA using High-Capacity cDNA Reverse Transcription Kit (Applied Biosystems). qRT-PCR was performed on a CFX96 Real-Time System (Bio-Rad) using the iTaq Universal SYBR Green Supermix (Bio-Rad). Expression levels were normalized to *Hprt* as a housekeeping gene (IDT, Leuven, Belgium; primer sequences are listed in Table 1).

### Cell culture

C2 myoblasts were cultured in DMEM supplemented with 15% FCS, 2% sodium pyruvate, 1% nonessential amino acids, 100 U/ml penicillin, and 0.1 mg/ml streptomycin (P/S). For all nutrient restimulation experiments, dialyzed FCS (10 kD cutoff; Invitrogen/Life Technologies) was used. Cells were deprived for the respective nutrient for 2 h and restimulated with full medium for another 2 h. HEK293T cells were cultured in DMEM containing 10% FCS and P/S. A7r5 cells were cultured in DMEM without phenol red and low glucose, supplemented with 10% FCS, P/S, and 2% L-glutamine (all cell culture components from PAN-Biotech).

### CRISPR/Cas9 knockout of *Cyth2* and rescue

Knockout cell lines were generated using the CRISPR/Cas9 system combined with lentiviral transduction as described before (Joung et al, 2017). gRNAs were used as published in the Human and Mouse CRISPR Knockout Pooled Library (Brunello and Brie library; Table 2 [Doench et al, 2016]) for gene editing. gRNA oligonucleotides were cloned into the pLentiCRISPRv2 vector (transfer plasmid, #52961; Addgene) using BsmBI restriction and Golden Gate assembly. gRNA-containing pLentiCRISPRv2 plasmids were cotransfected with the pMD2.G envelope plasmid (#12259; Addgene) and psPAX2 packaging plasmid (#12260; Addgene) into HEK293T cells using the calcium phosphate transfection method and incubated for 48 h to produce lentivirus particles. The filtered supernatant (0.45 µm pore size) was supplemented with 8 µg/ml polybrene and added to the target cell lines for 24 h. After an additional 24 h, transduced cells were selected with puromycin for 3 d, followed by subcloning (for C2 and HEK cells). Effective gene editing was confirmed by Western blot and Sanger sequencing for single-cell clones.

For rescue experiments, C2 myoblast clones were seeded on coverslips. After cells adhered, they were transfected with the appropriate plasmids (Table 3) with jetOPTIMUS (Polyplus Transfection). After 48 h, cells were fixed and stained for analysis of the Golgi apparatus.

### Western blot analysis and antibodies

Cells and organs from WT and *Cyth2*$^{-/-}$ mice were homogenized on ice in MRC lysis buffer (50 mM Tris–HCl, pH 7.5, 1 mM EGTA, 1 mM EDTA, 270 mM sucrose, 1% Triton X-100, protease and phosphatase

**Table 1. Primers for genotyping and qRT-PCR.**

| Target gene | Primer name | Sequence (5′-3′) |
|---|---|---|
| Genotyping | | |
| *Cyth2* (murine) | Pscd2 WT1 screen for | CAGAAATGCCAGGGCTTTCTCAGC |
| | Pscd2 WT1 screen rev | GCATAGGTTTCAGGGCTGGAAAACAC |
| | Pscd2 FRT screen rev | CGGAAGGAATGCCCAGCCAAAAT |
| Cre recombinase | CRE for | CCGGTCGATGGAGTGA |
| | CRE rev | GGCCCAAATGTGGATA |
| qRT-PCR | | |
| mouse *G6pc1* | mG6pc1 for | AGC TGA ACG TCT GTC TGT CC |
| | mG6pc1 rev | TTC TCC AAA GTC CAC AGG AG |
| mouse *Gck* | mGck for | GAA CAA CAT CGT GGG ACT TC |
| | mGck rev | AGC TCC ACA TTC TGC ATC TC |
| mouse *Hprt* | mHprt for | GCT GGT GAA AAG GAC CTC T |
| | mHprt rev | CAC AGG ACT AGA ACA CCT GC |
| mouse *Pklr* | mPklr for | CAT TGC TGT GAC TCG TTC TG |
| | mPklr rev | CAC AAT CAC CAG ATC ACC AA |
| mouse *Pck1* | mPck1 for | TGC CGG AAG AGG ACT TTG AG |
| | mPck1 rev | CAC TTG ATG AAC TCC CCA TC |

inhibitors). Western blots were performed as described previously (Mitschka et al, 2015). After blotting, nitrocellulose membranes were stained with Ponceau Red to determine equal loading of samples. Antibodies used were as follows: CYTH2 (clone H-7, 1:500; Santa Cruz), GAPDH (ACP001P, 1:10,000; Acris), pAKT (T308; #9275; 1:1,000), AKT (#9272; 1:1,000), p-p70S6K (T389; #9205; 1:1,000), p70S6K (#9202; 1:1,000), pULK1 (S757; #6888; 1:1,000), LC3b (#2775; 1:1,000) (all from Cell Signaling Technologies), and WFDC2 (PA5-80227; 1:1,000; Thermo Fisher Scientific).

## Lectin staining and flow cytometry

C2 myoblasts were harvested with 2 mM EDTA in PBS. Cells were stained with Alexa Fluor 488–conjugated lectins (Con A-Alexa 488 conjugate C11252; PNA-Alexa 488 conjugate L21409, WGA-Alexa 555 conjugate W32464; Thermo Fisher Scientific) and measured with a FACSCanto II. Analysis was done with FlowJo (v.10). To confirm the role of the Golgi apparatus in glycosylation, WT cells were treated with 5 μg/ml brefeldin A (Sigma-Aldrich) for 4 h before harvesting. Brefeldin A is an inhibitor of intracellular protein transport and therefore can induce cell death. Draq7 (BioLegend) was added (300 nM) 1 min before measurement to discriminate between live and dead cells.

## Differential centrifugation and mass spectrometry

The differential centrifugation to analyze subcellular protein localization was performed as described by Itzhak et al (2016). Briefly, C2 cells were harvested and washed once in ice-cold hypotonic lysis buffer (HLB, 25 mM Tris, pH 7.5, 50 mM sucrose, 0.2 mM EGTA, 0.5 mM MgCl$_2$, and protease inhibitors). Cells were lysed in HLB in a Dounce homogenizer with 20 strokes. The cell homogenate was immediately mixed with 10% vol/vol hypertonic sucrose buffer (25 mM Tris, pH 7.5, 2.5 M sucrose, 0.2 mM EGTA, 0.5 mM MgCl$_2$, and protease inhibitors) to restore the sucrose concentration to 250 mM. The cell lysate/supernatant was sequentially centrifuged at 1,000*g*, 10 min.; 3,000*g*, 10 min.; 5,500*g*, 15 min.; 12,200*g*, 20 min; 24,000*g*, 20 min.; 78,400*g*, 30 min. Each pellet was resuspended in RIPA buffer (150 mM NaCl, 1% Triton X, 0.5% NaDoc, 0.1% SDS, 50 mM Tris, pH 7.5). Samples were stored at −20°C until analysis.

Proteins enriched in organellar fractions and from total cell lysate were denatured using urea lysis buffer (6 M urea, 2 M thiourea, 10 mM Tris-(2-carboxyethyl)-phosphine, 30 mM 2-chloroacetamide in 50 mM Tris–HCl, pH 8.5). After incubation at RT for 30 min, urea concentrations were diluted to less than 1 M using 50 mM Tris–HCl. Proteins were enzymatically digested at RT with trypsin/Lys-C (Promega) at a 1:100 enzyme-to-protein ratio for 16 h. On the next day, the digestion was stopped by adding 10% formic acid to a final concentration of 1% and peptides were desalted using a modification of the stage-tip protocol (Rappsilber et al, 2007). In-house–made polystyrene-divinylbenzene reversed-phase sulfonate (SDB-RPS, Affinisep) double-layer stage tips were activated by the addition of 100 μl pure methanol. Tips were washed with 100 μl buffer B (80% acetonitrile, 0.1% formic acid in LC-MS-grade water) and equilibrated with 100 μl buffer A (0.1% formic acid in LC-MS-grade water). Digested peptides were loaded onto the stage tips with a volume equivalent to 20 μg of initial protein input material, and the following were desalted by one wash of 100 μl buffer A and two washes of 100 μl buffer B. Elution of peptides was achieved by the addition of 60 μl of buffer X (1% ammonia in acetonitrile). After each step, centrifugation was carried out at 800*g* for 2 min or until all liquid passed through the

**Table 2. sgRNAs for CRISPR knockout of *Cyth2*.**

| Target gene | Sequence (5′-3′) | sgRNA library |
|---|---|---|
| Mouse *Cyth2* | TTTGCAGTAAGACCTTGCAG | Brie |
| Human *CYTH2* | GCTCAGTGAAGCCATGAGCG | Brunello |

**Table 3. CYTH2-overexpression constructs for rescue experiments.**

| Vector backbone | Gene | Mutation | 5′Tag |
|---|---|---|---|
| pN1 | *Cyth2*-2G | None | RFP |
| pN1 | *Cyth2*-2G | E156K | RFP |
| pN1 | *Cyth2*-2G | DCC (2-46AS) | RFP |
| pN1 | *Cyth2*-3G | None | RFP |
| pN1 | *Cyth2*-3G | E156K | RFP |
| pN1 | *Cyth2*-3G | DCC (2-46AS) | RFP |

All inserts were verified by sequencing (Eurofins Genomics).

double layer. Finally, eluted peptides were vacuum-dried, and dissolved in buffer R (2% acetonitrile, 1% formic acid in LC-MS-grade water), peptide concentrations were determined, and 400 ng was used for injection into the LC-MS system.

Proteomics measurements were carried out with an ultrahigh-performance liquid chromatography EASY-nLC 1,200 system coupled online to an Orbitrap Exploris 480 tandem mass spectrometer (both from Thermo Fisher Scientific). Peptides were chromatographically separated on an in-house–produced analytical column (30 cm length, 75 $\mu$m inner diameter, 1.9 $\mu$m ReproSil-Pur 120 C18-AQ filling material [Dr Maisch]) and using a 90-min gradient consisting of buffer A and B. Starting at 4%, the amount of buffer B was linearly increased to 25% over 70 min at a flow of 300 nl/min. Buffer B was then linearly increased to 55% over 8 min and finally to 95% within 2 min. The analytical column was washed at 95% B for another 10 min.

Eluting peptides were online-transferred into the MS system by nanoelectrospray ionization operated at a constant voltage of 2.4 kV. Samples were analyzed in data-independent acquisition (DIA) mode. In brief, full MS spectra were recorded at a resolution of 120,000, a scan range of 380–1,020 m/z, an AGC target of 100%, and an injection time of 55 ms. DIA fragment spectra were recorded at a resolution of 15,000, an AGC target of 1,000%, and an injection time of 22 ms. In total, 75 DIA windows of 8 m/z window sizes were selected spanning a range from 400 to 1,000 m/z. Precursors were fragmented at an HCD collision energy of 31% and measured in centroid mode.

Mass spectrometry raw data were processed with the DIA-NN software tool (Demichev et al, 2020). A spectral library was predicted in silico from the UniProt SWISS-PROT *Mus musculus* database (version from 2021-11-18) with trypsin as the digesting enzyme, maximum number of missed cleavages set to 1, and cysteine carbamidomethylation enabled as a fixed modification. The scan window radius was set to 7, and mass accuracies were fixed to $2.08 \times 10^{-5}$ (MS2) and $3.6 \times 10^{-6}$ (MS1), respectively. Precursor masses were fixed to min 380 m/z and max 1,020 m/z with peptide sequence lengths from 7 to 30. Peptide-spectrum matches were filtered at an FDR < 0.01. Identified precursors were further filtered in R on Lib.Q.Values and Lib.PG.Q.Values < 0.01. Label-free quantified (LFQ) protein intensities were carried out with the maxLFQ algorithm implemented in the DIA-NN R package (Cox et al, 2014).

The statistical analysis was performed with the Perseus software suite (Tyanova et al, 2016) (v. 1.6.15). The following steps were performed consecutively to identify proteins that changed organellar localization upon protein knockout. Firstly, summed intensities were calculated for each protein within WT or knockout separately. Summed intensities were then subtracted individually from intensities in each fraction to get relative protein abundance. Delta abundance was calculated per fraction by subtracting KO relative abundance from WT relative abundance to reveal proteins changing organellar localization upon genotype. Only proteins present in each fraction of KO and WT were considered for multidimensional significance testing (threshold = 0.05, quantile = 0.55, iterations = 100, Benjamini–Hochberg FDR). Proteins with significantly different localizations were z-score–normalized and used for hierarchical clustering (Euclidean distance, clustering = k-means, 300 starting points, 10 iterations). Finally, enriched gene ontology terms enriched in individual clusters compared with all proteins were identified by Fisher's exact testing (Benjamini–Hochberg FDR = 0.05).

### Immunofluorescence and analysis with IMARIS

Cells were fixed with 4% paraformaldehyde, permeabilized and blocked in 2% BSA/0.2% Triton X-100/PBS for 30 min at RT, and incubated with the primary antibody (Golgin-97, #13192, 1:100; mTOR, #2983, 1:200; RAB5, #3547, 1:100; RAB7, #9367, 1:50; Cell Signaling; LAMP2, sc-18822, 1:100; Santa Cruz) in 1% BSA/0.1% Triton X-100/PBS for 1 h at RT. After subsequent incubation with Alexa Fluor 488–conjugated secondary antibodies in 0.05% Triton X-100/PBS for 45 min at RT, cells were mounted using Fluoroshield (ImmunoBioScience) containing 1 ng/$\mu$l DAPI. For Golgi visualization and mTOR-LAMP2 colocalization, z-stack images (0.1-$\mu$m intervals) were acquired with 63× magnification using an inverted, confocal laser scanning microscope (LSM 880+ Airyscan, Zeiss) and the ZEN software keeping imaging parameters constant for all slides.

Using IMARIS 9, microscopic images were analyzed to measure cell volumes and Golgi volumes, count RAB5-, RAB7-, and LAMP2-positive spots, and measure mTOR-LAMP2 colocalization.

Cells were analyzed using the cell detection algorithm as cell bodies with a single nucleus. Nuclei were detected as spots based on the DAPI signal with an estimated diameter of 8 $\mu$m. Cell bodies were detected based on the fluorescence of a surface marker protein. Automated detection of cells filtered for single nuclei per cell and separated cells by fluorescence declines at cell borders. Intensity thresholds were manually adjusted to catch every nucleus and entire cell bodies; partially imaged cells at the border of the detection area were omitted manually.

The surface detection algorithm was used to identify Golgi structures with a default surface grain size of 0.141 $\mu$m and automated background subtraction with default settings (0.529 $\mu$m diameter spherical subtraction). The intensity threshold was set manually. Detected surface structures smaller than 400 volume pixel ("voxel") were excluded automatically. Remaining surfaces

were grouped as a single Golgi stack, discarding incompletely captured structures, completely dispersed structures, and overlapping Golgi volumes from two neighboring cells.

RAB5-, RAB7-, and LAMP2-positive spots were analyzed using the spot detection algorithm. Default settings were applied apart from the estimated spot size, which was set to 0.5 $\mu$m in diameter. The detection intensity threshold was set to automatic thresholding to ensure an unbiased analysis of spots. Spots were matched with cell bodies to calculate the spot/cell volume ratio for each individual cell.

Colocalization of LAMP2 and mTOR was analyzed with the co-loc algorithm after baseline background subtraction. Pearson's correlation of intensities was taken as a measure for protein colocalization. For more details, see the IMARIS Reference Manual.

### Mass spectrometry–based proteomics of neonatal plasma

All chemicals were from Sigma-Aldrich unless otherwise noted (Sigma-Aldrich Chemie GmbH). Plasma samples were centrifuged for 1 min at 2,000$g$. Of the supernatant, 50 $\mu$g of protein per plasma sample was subjected to in-solution digestion with the iST 96x sample preparation kit (PreOmics GmbH) according to the manufacturer's recommendations (3-h digestion).

Peptides were separated with a Dionex UltiMate 3000 RSLCnano HPLC system (Dionex GmbH). 10 $\mu$g peptides were dissolved in 10 $\mu$l 0.1% formic acid (FA, solvent A), and 1 $\mu$l was injected onto an analytical column (400 mm length, 75 $\mu$m inner diameter, ReproSil-Pur 120 C18-AQ, 3 $\mu$m). Peptides were separated during a linear gradient from 5% to 35% solvent B (90% acetonitrile, 0.1% FA) at 300 nl/min over a 180 min. The LC was coupled to an Orbitrap Fusion Lumos mass spectrometer (Thermo Fisher Scientific). Data-independent acquisition was performed with the following scan parameters: 47 windows of 15 Da plus 0.5 Da overlap covering m/z 399.5–1,105.5. Isolated ions were fragmented with higher energy collision–induced dissociation (HCD) with 22%, 27%, and 32% stepped collision energy. Fragments were detected in the Orbitrap detector (profile-mode) with a resolution of 30,000 in the range of 200–1,800 m/z. AGC target was 500,000, and maximum injection time was 50 ms. Every 3 s, an MS1 scan was recorded in the range of 350–1,500 m/z, resolution = 120,000.

Data processing was performed with DIA-NN 1.8.1 (Demichev et al, 2020) in library-free mode based on the UniProt mouse reference proteome with isoforms (2023_03). The following parameters were applied: tryptic cleavage with one missed cleavage, variable modification of methionine by oxidation, acetylation of protein N terminus, static modification of cysteine by carbamidomethylation, output filtered at 1% FDR.

The statistical analyses of the DIA-NN precursor ion quantities were carried out in the R environment (R version 4.2.3) (R Core Team, 2018) using an in-house–developed workflow. Quantities with more than 65% missing values were removed. The data were variance-stabilized and transformed using the VSN package version 3.64.0 (Huber et al, 2002), and missing values were imputed using the method "v2-mnar" of the msImpute package version 1.6.0 (Hediyeh-Zadeh et al, 2023). The data were aggregated on the protein level using Tukey's median polish method. The statistical analysis to identify differentially abundant proteins was performed with the limma package version 3.52.4 (Ritchie et al, 2015). To account for correlations, present in the same litter of mice, litter was modeled as a random effect in the statistical analysis while using litter as a blocking parameter in the function duplicateCorrelation. The resulting *P*-values of the statistical contrast between the knockout and WT condition were adjusted for multiple testing, and the false discovery rates (FDR) were calculated by the Benjamini–Hochberg method.

### Tissue-specific *Cyth2*-deficient mice

Tissue-specific $Cyth2^{-/-}$ mice were generated by breeding $Cyth2^{+/-}$ mice with Flp-deleter mice (B6.SJL-Tg(ACTFLPe)9205Dym/J [Rodríguez et al, 2000]) to remove the transcriptional stop between two FRT sites behind exon 5. The following Cyth2 conditional mice were bred with nestin-Cre mice (brain-specific [Tronche et al, 1999]), myogenin-Cre mice (skeleton muscle-specific [Li et al, 2005]), alpha-myosin heavy chain mice (heart muscle-specific [Agah et al, 1997]), and albumin-Cre mice (liver-specific [Kellendonk et al, 2000]), respectively. Genotyping of adult mice was done from tissue of ear labeling.

### H&E staining of heart tissue

PFA-fixed paraffin-embedded murine heart cross-sections were deparaffinized by incubation in xylol twice for 5 min. Next, the sections were rehydrated by a descending ethanol dilution series (100% [2x, 3 min], and 96%, 70%, 50% [2 min each]), and dipped in distilled water. Cross-sections were stained in hematoxylin for 1 min and washed in cold running tap water for 10 min. Afterward, the sections were counterstained with 0.5% eosin for 2 min, and excess dye was rinsed in cold running tap water. Cross-sections were dehydrated in an ascending ethanol dilution series (50%, 70%, 96%, 100% [2x]) and cleared in xylol twice for 2 min. Sections were then mounted with Entellan (Merck), dried under the fume hood, and imaged by bright-field microscopy using a Olympus SZX10 (Olympus).

### LysoTracker staining of C2 myoblasts

WT and Cyth2-deficient C2 myoblasts were seeded onto ibidi slides (35 mm ibiTreat) and incubated overnight under standard culture conditions. Afterward, the cells were treated with 250 nM Lyso-Tracker Green DND-26 (Thermo Fisher Scientific) for 45 min in fresh medium, followed by three washing steps in live-cell imaging solution (HBSS, 5% FCS, 20 mM Hepes, pH 7.4) and imaging with a laser scanning confocal microscope. At least 20 cells per condition were imaged, and analysis was performed using the ImageJ Analyze Particles module identifying spots of 0.2–1.5 $\mu$m diameter.

### Statistics

*T* test (n ≥ 6 and Gaussian distribution), Mann–Whitney test (n ≤ 5 or no Gaussian distribution), or one-way ANOVA was calculated with GraphPad Prism software. Values of *P* < 0.05 were considered significant. Results are given as the mean ± SD or in a scatter plot with individual data points.

## Data Availability

The mass spectrometry proteomics data have been deposited to the ProteomeXchange Consortium via the PRIDE (Perez-Riverol et al, 2025) partner repository with the dataset identifiers PXD063657 (plasma proteomics) and PXD073571 (organellar maps), respectively.

## Supplementary Information

## Acknowledgements

We thank members of the Kolanus and Meissner laboratories for technical help and useful discussions, and we thank Jörg Höhfeld for comments on the article. We would like to thank the Analytical Proteomics Core Facility and the *Core Unit* for *Bioinformatics* Data *Analysis* of the Medical Faculty at the University of Bonn, namely, Marc Sylvester, Andreas Buness, and Farhad Shakeri, for providing support and instrumentation funded by the Deutsche Forschungsgemeinschaft (DFG, German Research Foundation)—including Projektnummer 386936527. The work was funded by the Deutsche Forschungsgemeinschaft (DFG, German Research Foundation) under Germany's Excellence Strategy-EXC2151-390873048 to W Kolanus and DFG FOR 2743: Mechanical Stress Protection-388932620. Furthermore, we would like to thank the TRR237 for providing support and resources for the development of the spatial proteomics analysis, funded by the Deutsche Forschungsgemeinschaft (DFG, German Research Foundation)—Projektnummer 369799452. This article is subject to HHMI's Open Access to Publications policy. HHMI laboratory heads have previously granted a nonexclusive CC BY 4.0 license to the public and a sublicensable license to HHMI in their research articles. Pursuant to those licenses, the Author-Accepted Manuscript of this article can be made freely available under a CC BY 4.0 license immediately upon publication. This publication was supported by the Open Access Publication Fund of the University of Bonn.

### Author Contributions

C Küsters: conceptualization, supervision, investigation, methodology, and writing—original draft, review, and editing.
B Jux: conceptualization, supervision, funding acquisition, investigation, methodology, and writing—original draft, review, and editing.
F Shakeri: conceptualization, supervision, investigation, methodology, and writing—original draft, review, and editing.
S Kallabis: conceptualization, supervision, funding acquisition, investigation, methodology, and writing—original draft, review, and editing.
F Meissner: investigation, methodology, and writing—review and editing.
W Kolanus: conceptualization, supervision, funding acquisition, investigation, methodology, and writing—review and editing.

### Conflict of Interest Statement

The authors declare that they have no conflict of interest.

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
