## [Reviewer comments · Life Science Alliance]

Cytohesin-2 is essential for the perinatal development of mice and regulates Golgi volume

Carsten Kuesters, Bettina Jux, Farhad Shakeri, Sebastian Kallabis, Felix Meissner, and Waldemar Kolanus
DOI: <https://doi.org/10.26508/lsa.202503429>

Corresponding author(s): Waldemar Kolanus, University of Bonn

Review Timeline:	Submission Date:	2025-06-20
	Editorial Decision:	2025-08-08
	Revision Received:	2025-12-17
	Editorial Decision:	2026-01-20
	Revision Received:	2026-01-27
	Accepted:	2026-01-29

Scientific Editor: Sarita Hebbar

Transaction Report:

August 8, 2025

Re: Life Science Alliance manuscript #LSA-2025-03429

Prof. Waldemar Kolanus
University of Bonn
LIMES Program Unit Molecular Immune and Cell Biology, Laboratory of Molecular Immunology
Carl-Troll-Straße 31, Bonn
Bonn, NRW 53115
Germany

Dear Dr. Kolanus,

Thank you for submitting your manuscript entitled "Cythoesin-2 is essential for the perinatal development of mice and regulates Golgi volume" to Life Science Alliance.

Your manuscript was assessed by three expert whose comments are appended below. As you will note that reviewers found your work of potential significance. However, they have also raised important concerns regarding all sections of the manuscript.

We concur with the reviewers that a revised manuscript must state and explain connections between the different phenotypes related to Cyth2 mutant.

For the amino acid sensing phenotype, we agree with that you should present the quantified data as the ratio of phospho-ULK1 to total ULK1, and that you should provide higher magnification inserts for Figure 3B.

For the golgi-related phenotypes, we concur with the reviewers that you should provide experimental evidence for localisation of Cyth2-tagged and mutant constructs (reviewers 1, 2), and for any kind of defect in Golgi trafficking in Cyth2 mutants as suggested by reviewer 3. We also agree that you should provide images to demonstrate any effect of transfection on Golgi architecture.

We encourage you to follow the reviewers' request for the following: analysed proteomics data and images of uncropped western blot data/Ponceau staining as additional supplemental information. Please also ensure that all blots are aligned properly and that any crop lines are indicated.

With this advice, we invite you to submit a revised manuscript addressing the reviewer comments. When submitting the revision, please include a letter addressing the reviewers' comments point by point. While a rebuttal must respond to all points in some form, additional experiments to resolve these points, other than indicated above, will not be required.

Thank you for this interesting contribution to Life Science Alliance. We are looking forward to receiving your revised manuscript.

Sincerely,

Sarita Hebbar, PhD
Scientific Editor
Life Science Alliance
<http://www.lsjournal.org>

B. MANUSCRIPT ORGANIZATION AND FORMATTING:

Reviewer #1 (Comments to the Authors (Required)):

LSA-2025-03429 Küsters et al.

Küsters and colleagues present a manuscript where they studied the role of cytohesin-2 (CYTH2), a GEF for Arf1 and Arf6, in embryonic development and in Golgi structure. They show that Cyth2 knock-out is lethal shortly after birth and proteomics analysis points to a role for Cyth2 in Golgi structure and secretion. The manuscript is interesting but lacks a bit coherence and suffers from several minor and major insufficiencies. Specifically, the following points need to be addressed before considering publication.

1. First chapter of results section, conclusion. What about a concluding sentence about all the different tissue-specific KOs, what do the results suggest?
2. Why were myoblasts selected for further studies, since the conditional KO in muscles did not show a phenotype?
3. Fig. 3A and other Western Blots. The individual blots are clearly not from the same membrane (different running behavior), so the Ponceau staining can only serve as loading control for one or some of them. This has to be clearly indicated! In Fig. 6G, brain samples, it seems as if the Ponceau does not fit to any of the Western Blots. Suppl. Fig. 3D was spliced, that has to be indicated more clearly, for example with a white or black line and a sentence in the legend. I suggest to add a figure with full size blots and all Ponceau stainings as supplement.
4. Fig. 3B is too small to see anything. Please show a higher magnification of representative inserts, ideally not in a merged color image but with the channels separated.
5. Chapter "Organelle proteomics maps..." The proteomic data should be made available as supplemental tables, so that interested readers can see for example which 177 were displaced, and which Golgi proteins are displaced. And the data should be deposited in a public repository.
6. "Employing organellar maps, we were able to confirm, that CYTH2 is localized at endosomes and the plasma membrane, but we also found CYTH2 enriched in protein complexes (Fig. 4E)." I don't understand how you come to that conclusion. Fig. 4E shows the relative distribution of CYTH2 over the fractions of the gradient, but that distribution is totally different from the distribution of endosomal and plasma membrane proteins in Fig. 4D (mostly in 5K and 12K). The legend of Fig. 4F is missing, as is its description in the text.
7. Suppl. Figs 3 E,F are too small.

8. Fig. 5 F,G. In my experience the Golgi often fragments in transient transfection, maybe because of the stress of the transfection as such or because of the transfection reagent. Don't you have that problem? Can you show representative images?

9. Why did you not make a GFP or RFP fusion protein to see the localization, of course keeping in mind that the tag can influence the localization?

10. Page 11/Fig. 6A. Why is the effect of Cyth2-KO stronger than BFA, at least for ConA and PNA? How long was the BFA incubation? BFA results in the complete blockade of any transport and consequently cell death after a while. I doubt that the CYTH2-KO cells are that severely affected, after all they seem to grow quite normal.

11. Discussion, "the range of studies...." Please cite those studies, after all this is the discussion section!

11. Discussion, same sentence. This is one of the coherences I meant. Your results and the work of others suggest that a main function of CYTH2 is in endosomes, but then all your follow-up studies concentrate on the Golgi? Can you explain that? And could the Golgi effects be a consequence of endosomal defects caused by CYTH2? Reduced flow to the Golgi, for example?

12. Suppl. Fig. 5. There is C and D in the figure, but not in the legend?

Reviewer #2 (Comments to the Authors (Required)):

Review of 'Cytohesin-2 is essential for the perinatal development of mice and regulates'
Carsten Küsters,#, Bettina Jux,#, Farhad Shakeri, Sebastian Kalabis, Felix Meissner, Waldemar Kolanus

Summary

This manuscript describes a role of small Arf GEF protein, Cytohesin-2 (CYTH2), in perinatal development and its critical function in maintaining Golgi structure and function. While the importance of Arf1 function as well as the role of big GEFs (GBP1 and BIG1/2) is better understood, the role of Cytohesins is less well understood. The GEF activity of CYTH2 has been previously reported for Arf1/6 function in endocytosis. While CYTH2 has been previously reported to also localize to the Golgi, its function at the Golgi is not characterized.

The authors generated both in vivo (cyth2^{-/-} mice) and in vitro (cyth2 KO C2 myoblast, HEK293T, A7r5 cell lines) models to study the phenotypes associated with cyth2 deficiency. They find that cyth2 KO leads to neonatal lethality, with almost no viable mice at P21. The tissue specific KO did not phenocopy the whole animal KO suggesting the lethality is due to systemic requirement of CYTH2 function. Using organellar proteomics, the authors report that several Golgi proteins are displaced in the absence of cyth2 function. Cyth2 KO has altered glycosylation, reduced Golgi volume, and secretion defects for plasma proteins such as WFDC2. Surprisingly, while Golgi volume defect can be rescued by CYTH2 overexpression, the GEF-inactive (E156K) mutant was also capable of restoring Golgi volume, indicating a GEF-independent mechanism for CYTH2's role in Golgi volume regulation.

Comments:

The manuscript convincingly establishes the physiological importance of CYTH2 and an effect of CYTH2 on Golgi structure and function, although it is unclear if the neonate lethality is specifically due a loss of its Golgi specific role or due to general defects in endocytosis. The work also provides evidence of defects in Golgi structure in cyth2 KO cell lines, but the authors do not provide direct evidence of CYTH2 at the Golgi. Their organelle fractionation MS experiments indicate CYTH2 at the endosomes, plasma membrane and protein complexes- but the identity of CYTH2 at the Golgi itself is unclear. Additionally, while the GEF mutant can correct the reduced Golgi volume, the molecular mechanism by which CYTH2 influences Golgi structure (e.g., which proteins it interacts with via its coiled-coil domain to maintain Golgi function) is unclear. Addressing the nuances of the proposed GEF-independent mechanism and providing further mechanistic clarity on the precise cause of neonatal lethality would further strengthen the impact of these findings.

Specific points:

1. What is the sub-cellular localization of RFP tagged CYTH2 and cyth2 mutants used in the work and what percent of the signal is contributed by their Golgi localization?
2. The authors show a couple of proteins that are displaced in the organelle proteomics exp, it would be useful to provide a list of all proteins (Golgi and other organelles) that are displaced in the cyth2 KO. Are there trends in this data? For instance, are most glycosyltransferases displaced or only a selected few mentioned in the text?
3. What are the interactors of the coiled coil domain of CYTH2? The authors speculate an adaptor like role for CYTH2 at the Golgi- for which proteins and how? For instance, does CYTH2 directly interact with C1GALT1 via its coiled coil domain to ensure Golgi localization of C1GALT1?
4. Do the CYTH2 domain mutants, especially the coiled coil domain mutant, affect protein stability as assessed by a western blot?
5. Minor comment: a typo on page 6, ULK1 is listed at UK1, and on page 7, listed as UILK1.

Reviewer #3 (Comments to the Authors (Required)):

This manuscript by Carsten Küsters et al., reports that Cytohesin-2 is identified as an essential regulator of perinatal survival and Golgi integrity. Full genetic ablation of cytohesin-2 in mice induces perinatal lethality within 20 hours postpartum, establishing its non-redundant role in early development. Comparative organellar proteomics revealed significant structural alterations in the Golgi apparatus of CRISPR/Cas9-generated cytohesin-2-deficient myoblasts, with immunofluorescence quantification demonstrating a marked reduction in Golgi volume, a phenotype rescued by cytohesin-2 reconstitution. Functional analyses further indicated impaired glycosylation capacity, evidenced by decreased galactose/N-acetyl-galactosamine residues via lectin staining, alongside reduced global protein secretion in neonatal knockout mice quantified by mass spectrometry. Based on these findings, the authors propose that cytohesin-2 critically maintains Golgi architecture and secretory functions necessary for perinatal viability.

While the findings are intriguing, the study lacks sufficient evidence to fully support its major conclusions. Additionally, the results section contains several disjointed parts that are not clearly linked. Below are the major and minor points the authors should address to strengthen their findings.

Major Points

1. Cohesion of Results Section: The Results section currently presents several findings that are not clearly connected. For instance, the decreased ULK1 phosphorylation in *Cyth2*^{-/-} myoblasts upon amino acid stimulation is not mentioned in the Abstract, and its relationship to altered Golgi volume remains unclear. Please clarify how these findings relate to each other and highlight key mechanistic links, both in the Abstract and throughout the Results.
2. Serum Amino Acid Levels (Fig 2C): The reported increase in serum amino acid levels in *Cyth2*^{-/-} mice contrasts with established findings, where impaired autophagy/mTOR signaling typically results in decreased serum amino acids. The conclusion that this increase reflects "poor amino acid sensing and uptake" needs a more thorough explanation. Please provide a detailed mechanistic discussion to reconcile this apparent discrepancy and justify your interpretation.
3. ULK1 Phosphorylation Quantification (Fig 3A): The statement regarding reduced ULK1 phosphorylation should be based on the ratio of phospho-ULK1 to total ULK1, since changes in total ULK1 could confound interpretation. Please revise the quantification accordingly and present both representative blots and quantification data.
4. mTOR Lysosomal Localization (Fig 3B): The difference in mTOR lysosomal localization is subtle and not clearly discernible in the current images. Please provide higher-magnification and rescue experiments to substantiate this observation.
5. Amino Acid Sensing (Fig 3C):
The inference of reduced amino acid sensing in *Cyth2*-deficient cells, based solely on mTOR pathway activity, is indirect. To strengthen this claim, please perform direct functional assays measuring amino acid sensing in these cells.
6. Golgi Morphology and Trafficking (Fig 4): Altered Golgi morphology in *CYTH2*-deficient cells does not demonstrate a defect in Golgi-derived vesicle trafficking. Please provide functional evidence for a trafficking defect, such as assays of cargo secretion kinetics, coat protein assembly, or cargo exit from the Golgi.
7. Experimental Validation of Predictions (Fig 6D): The authors need to experimentally validate key predictions from your data, particularly SLC7A5 expression or function, using commercially available antibodies.
8. Discussion Structure and Content: Please revise the Discussion for greater conciseness and coherence. Clearly articulate the importance of *CYTH2* in maintaining Golgi volume and discuss mechanistically how changes in Golgi volume could impact protein glycosylation.

Minor Points

1. S6K Nomenclature: Spell out the full name of S6K at first mention and briefly describe its role within mTOR signaling.
2. *Cyth2* KO Validation (Fig 3A): Include representative Western blot images confirming successful *Cyth2* knockout in myoblasts.
3. Western Blot Annotation: Clearly label molecular weight markers on all Western blot images.
4. Quantification Methods: Provide a detailed description in the Methods section of how Golgi size/volume was quantified using IMARIS software.
5. Figure Panel References: Correct all figure panel references to the appropriate format (e.g., "Fig 5E, 2G, and 3G" instead of "G2 and G3").
6. Image Presentation (Fig 5F & 5G): Include representative images along with quantification graphs for the experiments shown in these panels.

We thank all reviewers for their constructive comments on our manuscript. All their specific points are individually answered below. We also opted to change the overall structure of the manuscript to solve issues that seems to have caused confusion among some of the reviewers (see below, also see Cover letter for more explanation). We strongly believe that this has resulted in much improved clarity of the manuscript.

Point-to-point reply

Reviewer #1:

1. First chapter of results section, conclusion. What about a concluding sentence about all the different tissue-specific KOs, what do the results suggest?

Done as suggested. We added the summarized observations from the tissue-specific knockout. **Line 208-224**, Results, added: "As metabolic changes were not severe and since high plasma amino acid levels are not known to be severely detrimental, we consider these minor deficiencies unlikely causes of neonatal death of *Cyth2*-deficient mice. Attempting to phenocopy the neonatal lethality of *Cyth2*-deficient mice, we observed increased expression of gluconeogenesis genes in hepatic tissue to be the only similarity between full knockout and liver-specific knockout mice. Since we could not reproduce the full knockout phenotype with any of the various tissue-specific knockout mice tested, we speculate that *Cyth2* might play a systemic, organism-wide role in early postnatal development."

2. Why were myoblasts selected for further studies, since the conditional KO in muscles did not show a phenotype?

No single cell line could have been chosen to resemble any murine phenotype we observed in this study. However, we focused on the C2 myoblasts as a cell line which was established, easy to maintain and transfect, and most importantly not transformed. Even though not tested, we expected these cells to have fewer genomic aberrations and cancerous features compared to many standard cell lines used in laboratories world-wide, such as HeLa and HEK 293 cells. That way, we aimed for more physiological mTOR responses towards nutrient or growth factor restrictions, as tumorous cell lines often show aberrant mTOR activity.

3. Fig. 3A and other Western Blots. The individual blots are clearly not from the same membrane (different running behavior), so the Ponceau staining can only serve as loading control for one or some of them. This has to be clearly indicated! In Fig. 6G, brain samples, it seems as if the Ponceau does not fit to any of the Western Blots. Suppl. Fig. 3D was spliced, that has to be indicated more clearly, for example with a white or black line and a sentence in the legend. I suggest to add a figure with full size blots and all Ponceau stainings as supplement.

All Western Blots were revised and figure legends adapted to indicate the representative role of the shown Ponceau image, as suggested.

Images of full-size membranes and blots are added as source data as listed below. The region of interest is indicated.

SourceDataForFigures4&5

SourceDataForFigureS1

SourceDataForFigureS4A

SourceDataForFigureS4B

SourceDataForFigureS4C

4. Fig. 3B is too small to see anything. Please show a higher magnification of representative inserts, ideally not in a merged color image but with the channels separated.

Done as requested, image resolution was also increased.

5. Chapter "Organelle proteomics maps...." The proteomic data should be made available as supplemental tables, so that interested readers can see for example which 177 were displaced, and which Golgi proteins are displaced. And the data should be deposited in a public repository.

As suggested, lists of proteins (Table 1 & 2) relevant for the manuscript are added as a supplement. Further, proteomic data are deposited and will be accessible online upon publication.

6. "Employing organellar maps, we were able to confirm, that CYTH2 is localized at endosomes and the plasma membrane, but we also found CYTH2 enriched in protein complexes (Fig. 4E)." I don't understand how you come to that conclusion. Fig. 4E shows the relative distribution of CYTH2 over the fractions of the gradient, but that distribution is totally different from the distribution of endosomal and plasma membrane proteins in Fig.

4D (mostly in 5K and 12K). The legend of Fig. 4F is missing, as is its description in the text.

A short description in the legend was added and the paragraph about the localization of CYTH2 was rephrased. Our data align with previous observations about the localization of CYTH2 and add to those results, as an enrichment especially in the 78 K fraction indicates a localization with larger protein complexes.

Line 265-270 (old line 235), Results, re-phrased: “Employing organellar maps, we were able to confirm, that CYTH2 is localized at endosomes and the plasma membrane, but we also found CYTH2 enriched in protein complexes (Fig. 3E).” TO “Employing organellar maps, we found CYTH2 in all five fractions, showing that CYTH2 can localize to endosomes and the plasma membrane, but we also found a higher proportion of CYTH2 precipitating with large protein complexes (Fig. 3F). We were further able to verify these mass spectrometric results by Western Blot. (Fig. 3G).

7. Suppl. Figs 3 E,F are too small.

Adjusted as requested.

8. Fig. 5 F,G. In my experience the Golgi often fragments in transient transfection, maybe because of the stress of the transfection as such or because of the transfection reagent. Don't you have that problem? Can you show representative images?

Figure 5 was adjusted and images of the stained Golgi apparatus were added for data acquired with C2 myoblasts.

Occasionally, we found dispersed Golgi structures, which were omitted from the analysis of Golgi volumes. However, as visualized by Golgin-97 staining, there was no obvious and generalized effect of the transfection agents tested and or used throughout the study (Jet Optimus).

Fig R1 shows WT and *Cyth2*-deficient C2 myoblasts transfected with different transfection reagents to express eGFP. Though the Golgi apparatus was also stained in green (using Alexa Fluor 488-coupled antibodies), the difference of Golgi structure to cytosolic eGFP is clearly visible.

The Golgi structure was not majorly affected by transfection with Lipofectamine,

Pectfect or JetOptimus. However, for Lipofectamine and PecFect there are obvious efficiency differences between WT and KO cells. Therefore, we decided to use JetOptimus for the experiments, as this method produced reliable and reproducible results for both genotypes.

Fig. R1: Testing different transfection reagents on C2 myoblasts.

WT and *Cyth2*-deficient C2 myoblasts were transfected with an EGFP expression vector using different transfection reagents (Lipofectamine, Pecfect, JetOptimus). After two days of culture, cells were fixed using PFA and stained for Golgin-97. Scale bar: 10 μ m.

9. Why did you not make a GFP or RFP fusion protein to see the localization, of course keeping in mind that the tag can influence the localization?

We performed the experiments presented in the manuscript Figure 5 (now Fig. 4) with RFP fusion proteins of CYTH2 and its mutant variants. The images show a broad and unspecific distribution of RFP, but also the CYTH2-RFP fusion proteins. An accumulation of WT CYTH2 and the cc-mutant at the plasma membrane is visible, which is minor in case of the EIK mutant. This condensation appears independent of the 2G and 3G isoform of CYTH2. Further, the main pool of constructs appears to remain distributed throughout the cytoplasm.

Further, an accumulation of CYTH2 at the Golgi apparatus is not obvious by fluorescence microscopy. The overlap of RFP fluorescence with Golgin-97 staining is negligible. From these data we can conclude that CYTH2 does not particularly accumulate at the Golgi apparatus but is also not excluded from that structure.

Fig R2 shows exemplary images of C2 myoblasts over-expressing the different constructs quantified in Fig. 5 G-H (now Fig. 4 G-H). They provide evidence, that transfections did not generally disturb Golgi structure and show, how cells were chosen for the analysis. The white arrows in the first figure panel (first row, first image) indicate exemplary cells analyzed for the Golgi phenotype. The dark yellow arrows (first row, images one, three, four, and five) indicate cells, which were excluded from the Golgi phenotype analysis in overexpression experiments, due to 1) no RFP construct expression, 2) very high RFP construct expression, often accompanied by diffuse background signal of Golgin-97, and 3) a completely dispersed Golgi apparatus occasionally occurring without a particular pattern in the cells.

Fig. R2: Distribution of CYTH2-RFP constructs and exemplary choice of cells for Golgi analysis.

C2 myoblasts, WT and *Cyth2* knockout, were transfected with the indicated constructs and

cultured for two days. After fixation, cells were stained for Golgin-97 and with the nuclear dye DAPI. Scale bar: 10 μ m.

In panel 1, row 1: white arrows indicate cells suitable for Golgi analysis, orange arrows indicate cells excluded from the analysis as described above.

10. Page 11/Fig. 6A. Why is the effect of *Cyth2*-KO stronger than BFA, at least for ConA and PNA? How long was the BFA incubation? BFA results in the complete blockade of any transport and consequently cell death after a while. I doubt that the *CYTH2*-KO cells are that severely affected, after all they seem to grow quite normal.

WT C2 myoblasts were treated with Brefeldin A as a positive control for diminished lectin staining for 4 h with 5 μ g/ml final concentration. We are aware that longer incubation periods or higher concentrations lead to cells death. We chose Brefeldin A as a positive control to show that a complete disruption of Golgi transport leads to a broad defect in glycosylation and thus, to a decreased staining with various lectins (significant reduction for ConA, PNA, and WGA). Accordingly, Brefeldin A disperses the Golgi apparatus completely, leading to a loss of structured Golgin-97 staining. In contrast, *Cyth2*-deficient cells have only decreased Golgi volumes and apparently a still structured Golgi apparatus and only levels of PNA-detected core 1 glycosylation (galactose/N-acetyl-galactosamine) were decreased.

Thus, the effect of *Cyth2*-deficiency is more distinct and appears slightly stronger compared to Brefeldin A treatment, the biological interpretation remains difficult, though. Are these cells more compromised than Golgi-disrupted cells losing a wider range of glycosylation patterns to a lesser degree? The focus of these experiments was to show that the effect of *Cyth2* appears to be rather specific for a smaller subset of Golgi-related transport processes, affecting C1GALT1 as well as its binding partners C1GALT1C1 and MGAT1 and therefore affecting only PNA-mediated lectin binding, not a broader range of glycosylation structures.

11. Discussion, "the range of studies...." Please cite those studies, after all this is the discussion section!

Done as requested. References have been added (Casanova 2007; Gillingham and Munro 2007; Humphreys et al. 2012; Hurtado-Lorenzo et al. 2006; Moreau

et al. 2012; Yi et al. 2022).”

11. Discussion, same sentence. This is one of the coherences I meant. Your results and the work of others suggest that a main function of CYTH2 is in endosomes, but then all your follow-up studies concentrate on the Golgi? Can you explain that? And could the Golgi effects be a consequence of endosomal defects caused by CYTH2? Reduced flow to the Golgi, for example?

We added a short explanation to the manuscript to add this idea for the readers. Based on our data we can only speculate what you suggested.

The literature describes the main function of CYTH2 in endosomal trafficking close to the plasma membrane. Also in the literature, changes of Golgi size are always associated with a change in material flow towards or from the Golgi apparatus (excluding total Golgi disrupting effects such as the deletion of GRASP55 and GRASP65). The transportation network around the Golgi apparatus is quite complex though, as material arrives from protein synthesis, the plasma membrane, endosomal compartments, and is transported between the cisternae of the Golgi stack.

Therefore, we can only assume that CYTH2 affects transport processes around the Golgi apparatus, that further affect the Golgi volume. Even though we cannot provide more inside in the mechanisms, the CYTH2-dependent transport processes or any interaction partners, we wanted to highlight the effect of CYTH2 on the Golgi apparatus morphologically and functionally, as a so far neglected role of the small Arf-GEF.

Manuscript was rewritten for clarification.

Line 443-447 (old line 408), Discussion, added: “The observed changes in Golgi volume could result from an increased material influx or decreased efflux from the Golgi, as size variation of this organelle is often caused by transportation defects (...). Cyth2, an in vitro activator of Arf1, could directly or indirectly affect Golgi transport processes.”

12. Suppl. Fig. 5. There is C and D in the figure, but not in the legend?

The figure legend was adjusted accordingly.

Reviewer #2:

1. What is the sub-cellular localization of RFP tagged CYTH2 and cyth2 mutants used in the work and what percent of the signal is contributed by their Golgi localization?

See Fig R2 (provided for the reviewers exclusively): As the localization of CYTH2-RFP construct suggest, CYTH2 does not accumulate at any specific site within the cell. Recruitment to the plasma membrane was described and is partially visible here, but the majority of the protein appears to be residing and staying in a cytosolic pool. There is some overlap with the Golgi volume, but only in so far, that we would conclude that there is neither accumulation at nor exclusion from the Golgi apparatus.

2. The authors show a couple of proteins that are displaced in the organelle proteomics exp, it would be useful to provide a list of all proteins (Golgi and other organelles) that are displaced in the cyth2 KO. Are there trends in this data? For instance, are most glycosyltransferases displaced or only a selected few mentioned in the text?

An unbiased GO-term analysis prompted us to analyze the Golgi apparatus in more detail. Proteins associated with the Golgi were enzymes rather than structural proteins. However, not all glycosyltransferases were affected and the displaced glycosyltransferases were not specific for a certain motive but would add a medial or core structure in the glycosylation process (such as C1GALT1). So, we were not able to connect or relate a certain process or group of proteins to the disturbed glycosylation pattern in order to potentially identify the cause of neonatal lethality in full knockout mice.

3. What are the interactors of the coiled coil domain of CYTH2? The authors speculate an adaptor like role for CYTH2 at the Golgi- for which proteins and how? For instance, does CYTH2 directly interact with C1GALT1 via its coiled coil domain to ensure Golgi localization of C1GALT1?

The coiled-coil domain is involved in protein-protein interactions and was described to be sufficient but also required for CYTH2 targeting to the Golgi apparatus. To the best of our knowledge, these interactions were not further investigated though.

We aimed to identify CYTH2 interaction partners by pulldown of FLAG-tagged

CYTH2 and subsequent mass spectrometry-based proteomics. This attempt did not show a specific enrichment of Golgi proteins or single Golgi-associated proteins appearing in the manuscript or in the list of proteins re-localized.

Of note, C1GALT1 is located within the Golgi apparatus, as is its chaperone C1galt1c1. Only the latter contains a predicted transmembrane domain and a short, 6-amino acid-long cytoplasmic tail which could potentially interact with CYTH2. Alpha fold predictions suggest no interaction though.

References:

- Hiester, K.G. and Santy, L.C. (2013) The cytohesin coiled-coil domain interacts with threonine 276 to control membrane association. *PLoS one* 8, e82084.
- Klarlund, J.K., Holik, J., Chawla, A., Park, J.G., Buxton, J. and Czech, M.P.(2001) Signaling complexes of the FERM domain-containing protein GRSP1 bound to ARF exchange factor GRP1. *The Journal of biological chemistry* 276, 40065–40070.
- Lee, S.Y. and Pohajdak, B. (2000) N-terminal targeting of guanine nucleotide exchange factors (GEF) for ADP ribosylation factors (ARF) to the Golgi. *Journal of cell science* 113 (Pt 11), 1883–1889.
- Mansour, M., Lee, S.Y. and Pohajdak, B. (2002) The N-terminal coiled coil domain of the cytohesin/ARNO family of guanine nucleotide exchange factors interacts with the scaffolding protein CASP. *The Journal of biological chemistry* 277, 32302–32309.

4. Do the CYTH2 domain mutants, especially the coiled coil domain mutant, affect protein stability as assessed by a western blot?

Stability or transfection efficiency for the different CYTH2 constructs were not analyzed in detail. However, irrespective of their origin, differing expression levels of the CYTH2 constructs were taken into account when performing the microscopic analysis by choosing cells with comparable and “average” intensities for RFP. While this choice is fairly subjective, it ensured comparable expression levels of the different constructs on a single-cell level for the given time point (fixation 48 h post transfection).

5. Minor comment: a typo on page 6, ULK1 is listed as UK1, and on page 7, listed as UILK1.

These typos were corrected.

Reviewer #3:

Major Points

1. Cohesion of Results Section: The Results section currently presents several findings that are not clearly connected. For instance, the decreased ULK1 phosphorylation in *Cyth2*^{-/-} myoblasts upon amino acid stimulation is not mentioned in the Abstract, and its relationship to altered Golgi volume remains unclear. Please clarify how these findings relate to each other and highlight key mechanistic links, both in the Abstract and throughout the Results.

We considered the reviewer's insight and interest and addressed the major cohesion issues in the introductory paragraph of this reply and by re-phrasing the manuscript as listed above.

Further, to answer the specific question for the link between ULK1 and the altered Golgi volume: We do not want to claim a clear link between CYTH2-mediated transport, Golgi volume and ULK1 activity. We made these observations and wanted to document our findings. On this particular process we can only speculate that CYTH2 might influence Golgi architecture and function by regulating transport processes from or towards the Golgi apparatus. A functional alteration of the Golgi could trigger organellar stress responses and besides its major role in regulating macroautophagy, ULK1 is also involved in organelle homeostasis as the inducer of organelle-specific autophagy. Further, ULK1 was shown to be involved in the regulation of ER-to-Golgi trafficking (<http://dx.doi.org/10.1016/j.molcel.2016.04.020>). That way, ULK1 could try to compensate the trafficking defect caused by loss of CYTH2, being occupied and not available for phosphorylation by mTOR. Thus, ULK1 phosphorylation would rather serve as a marker for the defect in *Cyth2* knockout cells.

2. Serum Amino Acid Levels (Fig 2C): The reported increase in serum amino acid levels in *Cyth2*^{-/-} mice contrasts with established findings, where impaired autophagy/mTOR signaling typically results in decreased serum amino acids. The conclusion that this increase reflects "poor amino acid sensing and uptake" needs a more thorough explanation. Please provide a detailed mechanistic discussion to reconcile this apparent discrepancy and justify your interpretation.

We adapted the manuscript as described above and took emphasis from the metabolic link between increased amino acids and mTOR—mediated ULK1

phosphorylation as well as the transport of amino acid transporters. However, we also want to take the opportunity to give a more extensive explanation here:

The literature about the role of mTOR signaling and autophagy in the early postnatal period usually states decreased amino acid levels upon impaired autophagy or overshooting mTOR signaling. Defects in autophagy are believed to block the production of amino acids from proteins especially in cardiac and muscle tissue, which was shown to be crucial for postnatal survival, as these amino acids provide a nutrient source before nutrition from food intake takes over and suffices to meet the organism's nutrient demands. Increased mTOR signaling in that period blocks autophagy and thereby deprives the organism of free amino acids for energy consumption, as it communicates nutrient availability.

Consequently, an increased amount of free amino acids as detected in *Cyth2*-deficient newborn mice could indicate either an increased production of free amino acids (higher autophagic protein digestions due to aberrant autophagy or diminished mTOR signaling as a major break on autophagy via ULK1), or as a reduced consumption of free amino acids (lower cellular uptake rates from the blood stream; lower protein synthesis activity in turn due to decreased mTOR signaling; lower energy consumption due to increased availability of other energy sources).

3.ULK1 Phosphorylation Quantification (Fig 3A): The statement regarding reduced ULK1 phosphorylation should be based on the ratio of phospho-ULK1 to total ULK1, since changes in total ULK1 could confound interpretation. Please revise the quantification accordingly and present both representative blots and quantification data.

We adjusted Fig. 3A and Suppl. Fig4 (now combined in Suppl. Fig. 4), now showing the induction of the phospho-signal due to stimulation with amino acids, glucose, and FCS, respectively, based on the ratio of phospho-protein to total protein. The manuscript was rephrased accordingly.

As Reviewer 3 pointed out correctly, the ratio of phospho-signal to total protein could lead to a different interpretation of the data. The ULK1 blots show that total ULK1 levels are indeed lower in *Cyth2*-deficient cells compared to WT clones, but also slightly decline upon amino acid starvation. Likewise, phospho-ULK1

levels in starved cells (amino acid deprivation) were higher in Cyth2-deficient myoblasts compared to WT controls. So, when using the phospho-protein to total-protein ratio, we would argue that a relative interpretation between starved and re-fed condition is necessary.

Of note, ULK1 total protein levels are regulated based on its activity. Upon amino acid deprivation, mTOR signaling declines and pULK1(S757) levels drop, allowing the activation of ULK1 and the induction of autophagy. Active ULK1 is prone to be degraded as a negative feedback loop to shut down autophagy during prolonged absence of mTOR signaling (due to starvation, cellular stress etc.). Contemplating, how to display our data taking into account this regulatory circuit, we made a pragmatic decision: the phospho-protein/Ponceau levels needed the least calculation steps and therefore are as close as possible to the „raw“ data. For completeness, we added in the source data not only the original membranes and Western Blots, but also phospho-protein and total protein data separately.

4. mTOR Lysosomal Localization (Fig 3B): The difference in mTOR lysosomal localization is subtle and not clearly discernible in the current images. Please provide higher-magnification and rescue experiments to substantiate this observation.

Done as requested. The higher resolution images and the higher magnification inserts should suffice the requested clarification of a visible representation of the data within the plot. Rescue experiments were not planned, as the subtle effects would make a rescue difficult to prove.

5. Amino Acid Sensing (Fig 3C): The inference of reduced amino acid sensing in Cyth2-deficient cells, based solely on mTOR pathway activity, is indirect. To strengthen this claim, please perform direct functional assays measuring amino acid sensing in these cells.

To the best of our knowledge, direct amino acid sensing is difficult to detect. The cellular amino acid sensors described so far form a complex network of GEFs and GAPs and sequestering interaction partners of these, finally regulating the activity of the Rag-GTPases. Thus, GTP loading, click-chemistry-based amino acid binding, or direct protein interactions would be required to be detected. Cellular biosensors for that purpose are still in development.

All these assays are not only difficult to establish and calibrate, the complexity of the whole amino acid sensing would require a number of experiments way beyond the scope of this manuscript. The manuscript was revised to clarify the

focus of our study and the ULK1 data being an interesting, yet not fully understood observation coming up during our investigation. Whether the ULK1 observations are linked to the Golgi phenotypes or a separate process orchestrated by CYTH2 remains to be investigated and can only be speculated about here.

6. Golgi Morphology and Trafficking (Fig 4): Altered Golgi morphology in CYTH2-deficient cells does not demonstrate a defect in Golgi-derived vesicle trafficking. Please provide functional evidence for a trafficking defect, such as assays of cargo secretion kinetics, coat protein assembly, or cargo exit from the Golgi.

At this point, we cannot provide direct evidence for defective Golgi transport. The literature describes that Golgi morphology is mainly regulated by its structural components such as the GRASP proteins, and the material load within the Golgi, transported to or out of the organelle. We therefore speculate on a role of CYTH2 as a trafficking regulator, since a structural function of CYTH2 is highly unlikely. Evidence of defective transport is only provided indirectly, as secretion of proteins, a central role of the Golgi apparatus, seems overall impaired in *Cyth2*-deficient mice.

The manuscript was rewritten for clarification.

Line 443-447 (old line 408), Discussion, added: “The observed changes in Golgi volume could result from an decreased material influx or increased efflux from the Golgi, as size variation of this organelle is often caused by transportation defects (...). *Cyth2*, an in vitro activator of Arf1, could directly or indirectly affect Golgi transport processes.”

7. Experimental Validation of Predictions (Fig 6D): The authors need to experimentally validate key predictions from your data, particularly SLC7A5 expression or function, using commercially available antibodies.

We aimed to stain SLC7A5 in C2 myoblasts. However, the commercially available antibody tested was not suitable for a specific immunofluorescence staining in our hands. Therefore, we decided to leave this observation as a side note, even though a verified delocalization of a surface protein would have been an additional hint towards defects in the secretory pathway.

8. Discussion Structure and Content: Please revise the Discussion for greater conciseness and coherence. Clearly articulate the importance of CYTH2 in maintaining Golgi volume and discuss mechanistically how changes in Golgi volume could impact protein glycosylation.

The manuscript was re-phrased and re-structured as listed above and described in the introductory paragraph of this reply.

Minor Points

1.S6K Nomenclature: Spell out the full name of S6K at first mention and briefly describe its role within mTOR signaling.

The full name of p70S6K and a short functional description was added to the manuscript as requested.

Line 167-169 (old line 147), Results, added: "... (the ribosomal protein S6 kinase beta-1, also known as S6K; phosphorylates S6 ribosomal protein to induce protein synthesis)..."

2.Cyth2 KO Validation (Fig 3A): Include representative Western blot images confirming successful Cyth2 knockout in myoblasts.

Representative blots for the depicted data and the loss of CYTH2 are shown.

3.Western Blot Annotation: Clearly label molecular weight markers on all Western blot images.

Done as requested, molecular weight markers were added to all blots shown in the manuscript.

4.Quantification Methods: Provide a detailed description in the Methods section of how Golgi size/volume was quantified using IMARIS software.

The used algorithms and parameters for the analysis using IMARIS software are now included in the methods section of the manuscript.

5.Figure Panel References: Correct all figure panel references to the appropriate format (e.g., "Fig 5E, 2G, and 3G" instead of "G2 and G3").

All figure references were revised carefully and the punctuation was adapted for better distinction between figure reference and reference to the figure content. For clarification, as CYTH2 was expressed in its two different isoforms, which are abbreviated with „2G“ and „3G“ in the literature based on the exon splicing which

leads to a switch in the amino acid sequence of only a single glycine residue, in the mentioned figure reference this difference was specifically mentioned.

6. Image Presentation (Fig 5F & 5G) (now Fig. 4F-H): Include representative images along with quantification graphs for the experiments shown in these panels.

Representative images were added to Fig. 4 (former Fig. 5) to show the Golgin-97 staining in WT and knockout cells as well as in RFP construct-transfected cells.

Additional list of significantly changed passages, wordings or figures, to address the overall feedback from the Reviewers and the cohesion issues mentioned specifically by Reviewer 1 (comment 11) and Reviewer 3 (comment 1) – line numbers refer to the revised manuscript:

Line 47-49 (old line 39), Abstract, added/re-phrased: Employing mass spectrometry-based organellar proteomics for the cellular analysis of cytohesin-2 function, we discovered a markedly altered Golgi apparatus in Cytohesin-2-deficient C2 myoblasts.”

Line 134-135 (old line 120), Results, added: “To identify the pathology of the severe, lethal phenotype of *Cyth2*-deficient mice, we aimed to investigate the role of *Cyth2* in various organs.”

Line 170-171 (old line 148), Results, added: “Thus, the insulin receptor response in the absence of *Cyth2* is hampered, however the dysregulation appears at a different step of the signaling cascade compared to *Cyth3*^{-/-} mice.

Line 177-179 (old line 149), Results, re-phrased: “An upstream kinase of S6K is mTOR, which has previously been described as an essential sensor for growth factors, amino acids, and other metabolites (...).” TO “Downstream of AKT and upstream of S6K lies the kinase mTOR, which has previously been described as an essential sensor not only for growth factors, but also amino acids and other metabolites (...).”

Line 180-181 (old line 151), Results, added: “While insulin injections were not feasible with newborn mice, we still assessed the nutritional state of the *Cyth2*^{-/-} mice.

Line 186 (old line 155), Results, added: “glycolysis genes”

Line 190 (old line 159), Results, added: “It was described before that serum amino acids

Line 193-195 (old line 153), Results, added: “We therefore wondered, if CYTH2 activity is required for autophagy and amino acid supply in newborn mice.”

Line 198-206 (old line 166), Results, removed and replaced: “These findings hint at poor amino acid sensing and uptake in *Cyth2*-deficient animals.” TO “These observations do not support the notion of a significant autophagic defect in *Cyth2*-deficient mice or cells. Interestingly, when starved and re-supplemented with FCS or amino acids, *Cyth2*^{-/-} C2 myoblasts showed a significant reduction of mTOR-mediated induction of ULK1 phosphorylation, ULK1 being the central autophagy-inducing kinase (Fig. S4). Co-localization experiments for mTOR and LAMP2 for investigating the mTOR recruitment to the lysosomal surface support a minor alteration of mTOR signaling, as *Cyth2*-deficient cells showed a small but significant reduction in co-localization of LAMP2 and mTOR (Fig. S5).”

Line 226-230 (old line 206), Results, added: “The neonatal lethality of full knockout mice, which could not be phenocopied by conditional knockout mice, and the knowledge that *Cyth2* is ubiquitously expressed and implicated in vesicle transport (Casanova 2007) prompted us to asked whether a loss of *Cyth2* might hinder the correct transport and positioning of certain proteins such as growth factor receptors, the V-ATPase, or others.”

January 20, 2026

RE: Life Science Alliance Manuscript #LSA-2025-03429R

Prof. Waldemar Kolanus
University of Bonn
LIMES Program Unit Molecular Immune and Cell Biology, Laboratory of Molecular Immunology
Carl-Troll-Straße 31, Bonn
Bonn, NRW 53115
Germany

Dear Dr. Kolanus,

Thank you for submitting your revised manuscript entitled "Cytohesin-2 is essential for the perinatal development of mice and regulates Golgi volume" to LSA.

Your revised manuscript was reviewed by all the original reviewers. As you will note, the reviewers conclude that the revised manuscript has addressed their concerns. We agree with Reviewer 2 that you must revise the abstract to reflect that the GEF activity of CYTH2 is dispensable.

In line with the reviewers' evaluation, we would be happy to publish your paper in Life Science Alliance pending final revisions necessary to meet our formatting guidelines.

MANUSCRIPT ORGANIZATION AND FORMATTING:

To avoid unnecessary delays in the acceptance and publication of your paper, please read the following information carefully. Full guidelines are available on our Instructions for Authors page, <https://www.life-science-alliance.org/authors>

- Please include scale bar information in the figure legend for figure 1D, 4A, and 4F.
- LSA does not permit references to unpublished data. Please amend the sentence in Line 113 to refer solely to prior publications.
- Please conduct a thorough spell and grammar check of the manuscript document.
- Please provide antibody concentrations and for Draq7 in the methods section.
- Please include the sequence for reference gene (Hprt) in qPCR experiments in Table S1.
- Thank you for providing a statement on data availability. Please confirm that the link to the provided ID (PXD063657) is functional, and please clarify that it contains both datasets (plasma proteomics and organelle proteomics datasets).
- Please note that LSA does not allow a supplemental methods section. Supplementary methods, related tables and references must be incorporated in the main manuscript document.
- Please upload your main manuscript text as an editable doc file.
- Please add the X and Bluesky handles of your host institute/organisation, as well as your own and/or one of the authors, in our system.
- Figure S5 has only one panel; therefore, please remove the label A from the current figure and its legend.
- Please upload your Supplementary Tables in editable .doc or Excel format.
- Please add a callout for Figure S3A-E, G; S4A-C and S6A-G to your main manuscript text.
- Please be sure that the authorship listing and order is correct.

LSA encourages authors to provide a 30-60 second video where the study is briefly explained. We will use these videos on social media to promote the published paper and the presenting author (for examples, see <https://docs.google.com/document/d/1-UWCfbE4pGcDdcgzcmiuJl2XMBJnxKYeqRvLLrLSo8s/edit?usp=sharing>). Corresponding

or first-authors are welcome to submit the video. Please submit only one video per manuscript. The video can be emailed to contact@life-science-alliance.org

FINAL FILES:

The following items are required for acceptance.

The license to publish form must be signed before your manuscript can be sent to production. A link to the license to publish form will be available to the corresponding author only. Please take a moment to check your funder requirements.

Thank you for your attention to these final processing requirements. Please revise and format the manuscript and upload materials as soon as you are able.

Thank you for this interesting contribution to the literature. We look forward to publishing your paper in Life Science Alliance.

Sincerely,

Sarita Hebbar, PhD
Scientific Editor
Life Science Alliance
<http://www.lsjournal.org>

Reviewer #1 (Comments to the Authors (Required)):

The authors made a big effort to address all my comments, substantially revised the manuscript and provided additional data for reviewers only. I have no further comments and support acceptance.

Reviewer #2 (Comments to the Authors (Required)):

This manuscript describes a role of small Arf GEF protein, Cytohesin-2 (CYTH2), in perinatal development and its function in maintaining Golgi structure and function. The GEF activity of CYTH2 has been previously reported for Arf1/6 function in endocytosis.

The authors generated both in vivo (cyth2^{-/-} mice) and in vitro (cyth2 KO C2 myoblast, HEK293T, A7r5 cell lines) models to study the phenotypes associated with cyth2 deficiency. They find that cyth2 KO leads to neonatal lethality. The tissue specific KO did not phenocopy the whole animal KO suggesting the lethality is due to systemic requirement of CYTH2 function. Cyth2 KO has altered glycosylation, reduced Golgi volume, and secretion defects for plasma proteins such as WFDC2. Surprisingly, while Golgi volume defect can be rescued by CYTH2 overexpression, the GEF-inactive (E156K) mutant was also capable of restoring Golgi volume, indicating a GEF-independent mechanism for CYTH2's role in Golgi volume regulation.

While this work provides strong evidence that Golgi structure and function is altered in the absence of CYTH2 function, this is not dependent on its GEF function, and it is not clear if this effect is direct or indirect. In response to reviewer comments, the authors have clarified and toned down their over-arching conclusions in the discussion and results sections.

Before the manuscript is accepted for publication, the authors should edit the abstract to include these details- that the observed Golgi phenotypes could be direct or indirect and that the GEF activity of CYTH2 is dispensable.

Reviewer #3 (Comments to the Authors (Required)):

The authors have addressed all of my comments.

January 29, 2026

RE: Life Science Alliance Manuscript #LSA-2025-03429RR

Prof. Waldemar Kolanus
University of Bonn
LIMES Program Unit Molecular Immune and Cell Biology, Laboratory of Molecular Immunology
Carl-Troll-Straße 31, Bonn
Bonn, NRW 53115
Germany

Dear Dr. Kolanus,

Thank you for submitting your Research Article entitled "Cytohesin-2 is essential for the perinatal development of mice and regulates Golgi volume". It is a pleasure to let you know that your manuscript is now accepted for publication in Life Science Alliance. Congratulations on this interesting work.

Your manuscript will now progress through copyediting and proofing. At the proofs stage, please ensure to make the following changes in the text:

1. Please complete or correct the sentence (line 107), "We and others observed no or minimal phenotypical effects in Cyth1, Cyth3, and Cyth4 knockout mice without challenging (Jux et al. 2019; Yamauchi et al. 2012)."
2. Please remove "A)" from the legend of Figure S5.
3. Please include a callout for Figure S3G in the main manuscript text.

It is journal policy that authors provide original data upon request.

DISTRIBUTION OF MATERIALS:

Again, congratulations on a very nice paper. I hope you found the review process to be constructive and are pleased with how the manuscript was handled editorially. We look forward to future exciting submissions from your lab.

Sincerely,

Sarita Hebbar, PhD
Scientific Editor
Life Science Alliance
<http://www.lsajournal.org>